# No-regret Learning in Price Competitions under Consumer Reference Effects

**Negin Golrezaei**
MIT Sloan School of Management
golrezae@mit.edu

**Patrick Jaillet**
MIT Electrical Engineering and Computer Science
jaillet@mit.edu

**Jason Cheuk Nam Liang**
MIT Operations Research Center
jcnliang@mit.edu

## Abstract

We study long-run market stability for repeated price competitions between two firms, where consumer demand depends on firms' posted prices and consumers' price expectations called *reference prices*. Consumers' reference prices vary over time according to a memory-based dynamic, which is a weighted average of all historical prices. We focus on the setting where firms are not aware of demand functions and how reference prices are formed but have access to an oracle that provides a measure of consumers' responsiveness to the current posted prices. We show that if the firms run no-regret algorithms, in particular, online mirror descent (OMD), with decreasing step sizes, the market stabilizes in the sense that firms' prices and reference prices converge to a stable Nash Equilibrium (SNE). Interestingly, we also show that there exist constant step sizes under which the market stabilizes. We further characterize the rate of convergence to the SNE for both decreasing and constant OMD step sizes.

## 1 Introduction

In markets with repeated consumer-seller interactions, consumers develop price expectations (or reference prices) based on past observed prices. Such price memories would influence consumers' willingness-to-pay and hence their purchasing decisions, eventually impacting the overall aggregate market demand. Due to such memory dependent reference price effects, developing pricing strategies is challenging because firms may not necessarily know how consumers form and adjust price expectations. The complexity of pricing is further increased with competition, as competitors' pricing decisions impact not only a firm's immediate demand but also consumers' reference prices. Such challenges in pricing under competition and reference price effects make market stability particularly attractive to firms: under stable markets, long-term organizational planning and business strategy development can be conducted more effectively (see [14]). Inspired by this, in this paper, *we study the impact of consumer reference prices on the long-term stability of competitive markets.*

We examine a simplified market scenario where two firms sequentially set prices to sell goods over an infinite time horizon, and demand of each firm's goods are influenced by both firms' current prices and the current consumers' reference price, which is a weighted average of all past price trajectories. Also, the repeated price competitions occur in an opaque environment, where firms are not aware of any demand or reference price characteristics, and only have access to an oracle that returns consumers' responsiveness to posted prices.[1] In such a market scenario, we consider that both

firms run a general online mirror descent (OMD) algorithm.[2] Despite its simplicity, OMD algorithms have been theoretically shown to have good performance guaranties in both purely stochastic and adversarial environments (see [13, 52, 51]), and hence would be a plausible option for firms in this opaque environment of interest.

Our goal is to investigate whether firms' prices and consumer reference prices eventually stabilize in the long-run if firms run OMD. The notion of stability that we consider is represented by the convergence of firms' price profiles and reference prices such that there is no incentive for firms to deviate, eliminating the possibility for long-run price cycles and fluctuations. Similar notions of stability under dynamic competition has been studied under various equilibrium frameworks, and most relevant to this work are Markov perfect equilibrium and stationary equilibrium (see for example [27, 20, 18, 49, 2]). Nevertheless, these frameworks assume firms have complete information or optimize pricing decisions according to some prior on their competitors (or their aggregate) and the market. In contrast, our work focuses on competition in an opaque environment where firms post prices using no-regret learning algorithms like OMD. Here, we point out that our objective is not to present dynamic pricing polices that maximize firms' cumulative revenue. Instead, we seek to shed light on whether simple pricing polices like OMD that do not require a large amount of information eventually achieve market stability. Our contributions are summarized as follows.

- We characterize stability for dynamic competitive markets under consumers' reference price effects by defining the notion of *Stable Nash Equilibrium (SNE)*. We theoretically demonstrate its existence and shed light on its structural properties; see Theorem 3.1.

- We transform the two-firm game with a dynamic state (reference price) that varies in time according to firms' posted prices to a three-firm game without a state. The added virtual firm, which is referred to as nature, runs OMD with a constant step size (i.e. has a fast learning rate), and models how reference prices are affected by firms' past pricing decisions.

- We show that prices and reference prices converge to an SNE and achieve stable markets when the two (real) firms adopt decreasing step sizes that go to zero at a moderate rate; see Theorem 5.1 for details. We further show that with decreasing step sizes, the market stabilizes at a linear rate. We highlight that obtaining these convergence results is challenging because in our three-firm game, there is a firm (nature) who adopts a constant step size and learns at a fast rate. Our results show that despite the need to deal with such an inflexible virtual firm, the real firms can stabilize the market by adopting decreasing step sizes. In fact, the existence of the inflexible virtual firm in our game does not allow us to use the results in the literature on multi-agent online learning, where multiple interacting agents make sequential decisions via running the OMD algorithm to maximize individual rewards (see [34, 10, 35]). More specifically, in the multi-agent online learning literature, agents in the system of interest typically use step sizes of the same order (i.e. homogeneously decreasing or constant). In contrast, in our setting, while firms may take decreasing step sizes, the nature's step sizes is constant.

- Interestingly, we also show that there exist constant step sizes under which markets will converge to an SNE at much faster rates compared to adopting decreasing step sizes. Additionally, we show through an example that not every constant step size results in a stable market. Roughly speaking, if the firms' constant step size is compatible with nature's constant step size, the market stabilizes at a faster rate compared to decreasing step sizes; see Corollary 5.3.1 and Theorem 5.4 for details.

We refer the readers to Appendix A for an expanded literature review.

## 2   Preliminaries

**Consumer Demand and Reference Price Update Dynamics.**   We study a dynamic system where two firms simultaneously set prices in each period over an infinite time horizon to sell goods to consumers whose willingness-to-pay is affected by their price expectations, referred to as *reference prices*. We assume that the number of consumers is large so that demand for each firm is governed by the aggregate behavior of all consumers. Specifically, the demand of firm $i \in \{1, 2\}$ in time period $t$ with posted prices $\boldsymbol{p}_t = (p_{1,t}, p_{2,t})$ and consumers' reference price $r_t$ is given by

$$d_i(p_{i,t}, p_{-i,t}, r_t) = \alpha_i - \beta_i p_{i,t} + \delta_i p_{-i,t} + \gamma_i r_t \,, \tag{1}$$

where $p_{i,t}$ is the price of firm $i$ and $p_{-i,t}$ is the price of the other firm. To simplify notation, we may denote $d_i(p_{i,t}, p_{-i,t}, r_t)$ with $d_i(\boldsymbol{p}_t, r_t)$. We assume prices $p_{i,t}$ and reference prices $r_t$ are bounded, i.e., for $i \in \{1, 2\}$, $p_{i,t}, r_t \in \mathcal{P} = [\underline{p}, \bar{p}]$ for some $0 < \underline{p} < \bar{p} < \infty$, and $d_i(p_i, p_{-i}, r) \geq 0$ for any $p_i, p_{-i}, r \in \mathcal{P}$. The boundedness of prices corresponds to real-world price floors or price caps and is not unnatural. In Equation (1), $\alpha_i, \delta_i, \gamma_i > 0$ and $\beta_i \geq m(\delta_1 + \delta_2 + \max\{\gamma_1, \gamma_2\})$, where $m > 0$. Later in this section, we will provide an interpretation for these parameters that characterize our linear demand model. We note that linear demand models, which are widely used in the literature (see [29] for a comprehensive survey), can be viewed as a first-order approximation to more complex models.

After firms post prices, reference prices update according to the following dynamics:

$$r_{t+1} = ar_t + (1-a)(\theta_1 p_{1,t} + \theta_2 p_{2,t}), \tag{2}$$

where $\theta_1, \theta_2, a \in (0, 1)$ and $\theta_1 + \theta_2 = 1$. Here, $\theta_i$, which is independent of prices, represents how visible firm $i$ is to consumers: the larger the $\theta_i$, the more visible firm $i$ is, and the more it influences consumers' price expectations. The reference price update dynamics can be viewed as a memory-based process that characterizes how consumers adjust price expectations for goods over time as they observe new prices. Reference prices are formed by a weighted average of historical prices, where more recent prices are assigned larger weights. The specific exponential weighting scheme adopted in this paper has been motivated and empirically validated in the literature of behavioral economics (see, for example [50, 45, 25]). The parameter $a$ in the reference price update model characterizes to what extent consumers' reference price depends on past prices: As $a$ increases, the reference prices depend less on recently observed prices. Empirical estimates of $a$ typically range from $0.47$ to $0.925$ (see [25, 11]) depending on the type of goods sold.

We now provide an economic interpretation for our linear demand model by rearranging terms:

$$d_i(p_{i,t}, p_{-i,t}, r_t) = \alpha_i - (\beta_i - \gamma_i)p_{i,t} + \delta_i p_{-i,t} + \gamma_i(r_t - p_{i,t}). \tag{3}$$

When the posted price is greater than the reference price, i.e., $p_{i,t} > r_t$, the value $p_{i,t} - r_t$ can be viewed as the consumers' perceived price surcharge w.r.t. the reference price, and when $p_{i,t} < r_t$, the value $r_t - p_{i,t}$ is consumers' perceived price discount. Observe that in this rearrangement, demand increases when consumers' perceived price discount $(r_t - p_{i,t})\mathbb{I}\{r_t > p_{i,t}\}$ increases, and decreases as price surcharge $(p_{i,t} - r_t)\mathbb{I}\{p_{i,t} > r_t\}$ increases, which is a conventional representation of how reference prices affect consumer decisions in the related literature, see, for example, [42, 39]. Furthermore, the coefficients $\beta_i - \gamma_i, \delta_i$, and $\gamma_i$ measure the demand sensitivity of firm $i$ to its own prices $p_{i,t}$, its competitor's prices $p_{-i,t}$, and price surcharge/discount respectively.[3] With these interpretations, parameter $m > 0$ in the condition of $\beta_i \geq m(\delta_1 + \delta_2 + \max\{\gamma_1, \gamma_2\})$ can be viewed as a *sensitivity margin* that represents to what extent demand is more sensitive to a firm's own prices relative to competitor's prices and surcharge/discount. Take for example the case where $m = 1$: we have $\beta_i - \gamma_i > \delta_i + \delta_{-i}$, which means the impact of firm $i$'s price on its demand is greater than the aggregate impact of its price on the competitor's demand and the competitor's price on firm $i$'s demand (see Equation (3)). Additionally, for $m = 2$, we have $\max\{\gamma_1, \gamma_2\} < \beta_i - \gamma_i$, which represents the fact that reference effects in the market due to surcharge/discounts are generally less influential than any firm's price on its own demand.

We point out that the aforementioned relationships between model parameters $\{\alpha_i, \beta_i, \delta_i, \gamma_i, \theta_i\}_{i=1,2}$ lead to a diagonally dominant Jacobian matrix w.r.t. some mapping that characterizes the linear system consisting of firms and reference prices (particularly linearity in demand and reference price updates). We will provide further details on this particular mapping and its relevance with variational inequalities in Section 5.3.

**Market Stability.** In this work, our goal is to present simple pricing policies for the firms that stabilize the market even when firms do not have complete information on market conditions. Define $\pi_i(\boldsymbol{p}, r) := p_i \cdot d_i(\boldsymbol{p}, r)$ as the single-period firm $i$'s revenue when prices are $\boldsymbol{p}$ and the reference price is $r$. We say the market is stable at point $(\boldsymbol{p}^*, r^*)$ if the following two conditions hold:

1. **Best-response Conditions.** for $i \in \{1, 2\}$, we have $\pi_i(p_i^*, p_{-i}^*, r^*) \geq \pi_i(p, p_{-i}^*, r^*)$ for any $p$ in the feasible set $\mathcal{P}$; that is, firm $i$ cannot increase its revenue by posting another price $p \neq p_i^*$ when the other firm posts a price of $p_{-i}^*$ and the reference price is $r^*$.

2. **Stability Condition.** $r^* = \theta_1 p_1^* + \theta_2 p_2^*$; that is, the reference price does not change if the firm $i \in \{1, 2\}$ keeps posting price $p_i^*$; see Equation (2).

Throughout the paper, we may refer to a point $(\boldsymbol{p}^*, r^*)$ that satisfies the aforementioned conditions as a Stable Nash Equilibrium (SNE).

**Firms' Information Structure.** We present pricing policies under a partial information setting. In this setting, a firm $i$ does not know $d_i$, $d_{-i}$, reference price update dynamics, and does not observe any of historical competing prices nor the current reference price. To be more specific, in this setting, firms do not know the specific form of the demand functions and reference update dynamics, which in our case are linear. Nevertheless, we assume that after firms post prices $\boldsymbol{p}_t$ under reference price $r_t$, they can access a first-order oracle that outputs $\partial \pi_i(\boldsymbol{p}_t, r_t)/\partial p_i$, which intuitively represents consumers' responsiveness to a firm's prices under current market conditions.[4] We note that the partial information setting models real-world opaque environments where firms do not possess information of the market or its competitors. In this setting, firms set prices simultaneously, so a firm does not observe its competitor's pricing decision in the current period before setting its own price.

## 3 Existence and Structural Properties of SNE

In this section, we show that an SNE exists. Recall that for any SNE, each firm best responds to its competitor as well as consumers' reference price with no incentive for unilateral deviation. Let $\psi_i(p_{-i}, r) = \operatorname{argmax}_{p \in \mathcal{P}} \pi_i(p, p_{-i}, r)$, $i \in \{1, 2\}$,[5] be firm $i$'s best-response to the reference price $r$ and the price of the other firm $p_{-i}$. Further, for any reference price $r$, define set $\mathcal{B}(r)$ as follows

$$\mathcal{B}(r) = \{\boldsymbol{p} \ : \ p_i = \psi_i(p_{-i}, r), i = 1, 2\} . \tag{4}$$

As we will show in Theorem 3.1 below, $\mathcal{B}(r)$ is non-empty and when it is not a singleton, it is an ordered set with total ordering.[6] To show the existence of an SNE, we consider a simple pricing strategy that works as follows: in each period, firms set the largest best response profiles $\boldsymbol{p}_t$ w.r.t. reference price $r_t$, i.e., $\boldsymbol{p}_t = \max \mathcal{B}(r_t)$ (because $\mathcal{B}(\cdot)$ is an ordered set, $\max \mathcal{B}(r_t)$ is well-defined). We show that for any initial reference price $r_1 \in \mathcal{P}$, $(\boldsymbol{p}_t, r_t)$ converges monotonically to an SNE. Of course, this pricing strategy is only possible under the complete information setting, where each firm knows its own demand function $d_i$, its competitor's demand function $d_{-i}$, and the current reference price. That is, the described pricing strategy cannot be implemented in our partial information setting. Nevertheless, the convergence under this policy confirms the existence of an SNE.

**Theorem 3.1** (Existence of an SNE). *Let $\mathcal{B}(r)$, defined in Equation (4), be the set of best-response profiles w.r.t. reference price $r$. Then, for a fixed reference price $r \in \mathcal{P}$, $\mathcal{B}(r)$ is non-empty, and when $\mathcal{B}(r)$ is not a singleton, it is an ordered set with total ordering. Furthermore, assume that in each period $t$, firms set the largest best response prices $\boldsymbol{p}_t$ w.r.t. reference price $r_t$, i.e., $\boldsymbol{p}_t = \max \mathcal{B}(r_t)$. Then, for any initial reference price $r_1 \in \mathcal{P}$, $(\boldsymbol{p}_t, r_t)$ converges monotonically to an SNE.*

The proof of the first half of the result regarding the structural properties of the set of best response profiles $\mathcal{B}(r)$ is inspired by that of Tarski's fixed point theorem (e.g., see [19]). The proof of the second half regarding the convergence of the pricing policy builds on that of Theorem 6 in [36]. (This theorem shows the monotonocity of pure-strategy Nash Equilibrium for paramtererized games.) Detailed proofs can be found in Appendix B. Theorem 3.1 illustrates structural properties of SNEs: since $\mathcal{B}(\cdot)$ is an ordered set with total ordering, if there are multiple SNE's, any two SNE's $(\boldsymbol{p}_a^*, r_a^*)$ and $(\boldsymbol{p}_b^*, r_b^*)$ must either satisfy $\boldsymbol{p}_a^* \geq \boldsymbol{p}_b^*$ or $\boldsymbol{p}_a^* \leq \boldsymbol{p}_b^*$ under component-wise comparisons.

Due to the decision set boundaries, there may exist multiple SNE's. However, to simplify our analyses, in the rest of the paper we assume that there exists an SNE that lies within the interior of the action set $\mathcal{P}$. Under this assumption, Lemma 3.2 shows that the interior SNE is unique.

**Assumption 1.** *There exists an SNE $(\boldsymbol{p}^*, r^*)$ such that $(\boldsymbol{p}^*, r^*) \in (\underline{p}, \bar{p})^3$.*

**Lemma 3.2** (Uniqueness of SNE). *Under Assumption 1, there is a unique SNE $(\boldsymbol{p}^*, r^*) \in (\underline{p}, \bar{p})^3$.*

# 4 No-regret Pricing Policies under Partial Information Setting

Recall that under partial information, firms are unaware of the consumer demand function (they do not know the demand function is linear), reference prices, and reference price update dynamics. Hence, a natural approach for firms to increase revenue is to employ so-called *no-regret online learning* algorithms that adjusts prices in a dynamic fashion. We study the regime in which firms adopt the general OMD algorithm. We start by the following standard definition.

**Definition 4.1** (Strong convexity). Let $\mathcal{C} \subset \mathbb{R}$ be a convex set. A function $R : \mathcal{C} \to \mathbb{R}$ is said to be $\sigma$-strongly convex if for any $x, y \in \mathcal{C}$, we have $R(x) - R(y) \geq \frac{dR(y)}{dy}(x - y) + \frac{\sigma^2}{2}(y - x)^2$.

In the OMD algorithm, each firm $i$ chooses a continuously differentiable and strongly convex regularizer $R_i : \mathbb{R} \to \mathbb{R}$ associated with strong-convexity parameter $\sigma_i$, a sequence of step sizes $\{\epsilon_{i,t}\}_t$, and, for our convenience, minimizes the cost function (i.e. inverse of revenue) $\widetilde{\pi}_i := -\pi_i$, which is convex in $p_i$. Here, we assume each regularizer also satisfies a standard "reciprocity condition" used in optimization and online learning literature [15, 31, 5], i.e. whenever $x \to y$ for $x, y \in \mathbb{R}$ we have $D_i(x, y) \to 0$ where $D_i$ is the Bregman divergence w.r.t. $R_i$.[7] In OMD, each firm $i$ maintains a *proxy variable* $y_{i,t} \in \mathbb{R}$ over time, and in each period $t$, conducts pricing according to the following three steps:

1. Project the proxy variable $y_{i,t}$ back to the decision interval $\mathcal{P} = [\underline{p}, \bar{p}]$: $p_{i,t} = \Pi_{\mathcal{P}}(y_{i,t})$, where $\Pi_{\mathcal{P}} : \mathbb{R} \to \mathcal{P}$ is the projection operator such that $\Pi_{\mathcal{P}}(z) = z\mathbb{I}\{z \in \mathcal{P}\} + \underline{p}\mathbb{I}\{z < \underline{p}\} + \bar{p}\mathbb{I}\{z > \bar{p}\}$.

2. Access the first-order oracle $g_{i,t} := g_i(\boldsymbol{p}_t, r_t)$ defined by $g_i : \mathcal{P}^3 \to \mathbb{R}$, where

$$g_i(\boldsymbol{p}, r) = \partial \widetilde{\pi}_i(\boldsymbol{p}, r)/\partial p_i = 2\beta_i p_i - (\alpha_i + \delta_i p_{-i} + \gamma_i r) . \tag{5}$$

   This oracle can be viewed as a feedback mechanism that outputs the payoff gradient $\partial \widetilde{\pi}_i/\partial p_i$ evaluated at a given price profile $\boldsymbol{p}$ and reference price $r$. We note that the first-order feedback is very common in the optimization and learning literature as discussed in Section 2. Here, we point out that after a firm posts prices according to the OMD algorithm, it only obtains $g_{i,t}$, and does not necessarily observe the prices of its competitor nor the reference price.[8]

3. Update proxy variable $y_{i,t+1}$ such that $R_i'(y_{i,t+1}) = R_i'(p_{i,t}) - \epsilon_{i,t} g_{i,t}$,[9] where we define $R_i'(q) := \frac{dR_i(y)}{dy}\Big|_{y=q}$.

We summarize the two-firm OMD pricing scheme in Algorithm 1.

---

**Algorithm 1** 2-firm OMD pricing under reference price updates

**Input:** $\{R_i, \{\epsilon_{i,t}\}_t\}_{i=1,2}$, $y_{i,1} = \arg\min_{y \in \mathcal{P}} R_i(y)$ for $i = 1, 2$.
1: **for** $t = 1, 2, \ldots$ **do**
2:      **for** $i = 1, 2$ **do**
3:          Set price: $p_{i,t} = \Pi_{\mathcal{P}}(y_{i,t})$.
4:          Access gradient $g_{i,t} = g_i(\boldsymbol{p}_t, r_t)$.
5:          Update proxy variable:

$$R_i'(y_{i,t+1}) = R_i'(p_{i,t}) - \epsilon_{i,t} g_{i,t}.$$

6:      **end for**
7:      Reference price update (unobservable): $r_{t+1} = ar_t + (1 - a)(\theta_1 p_{1,t} + \theta_2 p_{2,t})$
8: **end for**

---

**Algorithm 2** Induced 3-firm OMD pricing with no reference price

**Input:** $\{R_i, \{\epsilon_{i,t}\}_t\}_{i=1,2,n}$, $y_{n,1} = r_1$, $y_{i,1} = \arg\min_{y \in \mathcal{P}} R_i(y)$ for $i = 1, 2$.
1: **for** $t = 1, 2, \ldots$ **do**
2:      **for** $i = 1, 2, n$ **do**
3:          Set price: $p_{i,t} = \Pi_{\mathcal{P}}(y_{i,t})$.
4:          Access gradient $g_{i,t} = g_i(\boldsymbol{p}_t, r_t)$.
5:          Update proxy variable:

$$R_i'(y_{i,t+1}) = R_i'(p_{i,t}) - \epsilon_{i,t} g_{i,t}.$$

6:      **end for**
7: **end for**

---

One can think of this sequential price competition with reference prices as a state-based dynamic game model where the reference price plays the role of an underlying state: each player (i.e., firm) has a continuous action space $\mathcal{P}$ and payoff function $\widetilde{\pi}_i$ that depends on all players' actions as well

as an underlying state variable $r_t$ that undergoes deterministic transitions. However, the view that we will adopt in the rest of the paper perceives reference prices $r_t$ as price decisions $p_{n,t} = r_t$ posted by a virtual firm which we refer to as nature and denote it by $n$. This is possible if, for any $\widetilde{\pi}_i, R_i, \{\epsilon_{i,t}\}_t$ $(i = 1, 2)$, we are able to construct a universal nature cost function $\widetilde{\pi}_n(p_1, p_2, p_n)$, strongly convex regularizer $R_n : \mathbb{R} \to \mathbb{R}$, and step size sequence $\{\epsilon_{n,t}\}_t$, such that when firms 1, 2 and nature independently run the OMD algorithm with their respective regularizers and step sizes (as summarized in Algorithm 2), the resulting price profiles $\{p_{1,t}, p_{2,t}, p_{n,t}\}_t$ recover the respective prices $\{\boldsymbol{p}_t, r_t\}_t$ of Algorithm 1. Here, note that $g_{n,t} = g_n(p_{1,t}, p_{2,t}, p_{n,t}) = \partial \widetilde{\pi}_n(p_{1,t}, p_{2,t}, p_{n,t})/\partial p_{n,t}$. The following Proposition 4.1 formalizes this view and shows that such $\widetilde{\pi}_n, R_n$, and $\epsilon_{n,t}$ indeed exist. The proof is provided in Appendix C, and we will refer to the dynamic game characterized in Algorithm 2 as the *induced 3-firm dynamic game*.

**Proposition 4.1** (Induced 3-firm dynamic game). *Fix any $\widetilde{\pi}_i, R_i, \{\epsilon_{i,t}\}_t$, $i = 1, 2$, and initial reference price $r_1$. If nature (called firm $n$) is associated with cost function $\widetilde{\pi}_n(\boldsymbol{p}, r) = \frac{1}{2}r^2 - (\theta_1 p_1 + \theta_2 p_2)\, r$, and chooses regularizer $R_n(r) = \frac{1}{2}r^2$ and step size $\epsilon_{n,t} = 1 - a$, for any $t \geq 1$, then the price profiles $\{p_{1,t}, p_{2,t}, p_{n,t}\}_{t\geq 1}$ resulting from the game in Algorithm 2 recovers the induced price and reference price trajectory $\{\boldsymbol{p}_t, r_t\}_{t\geq 1}$ of Algorithm 1.*

We note that the choices for nature's cost function $\widetilde{\pi}_n$, regularizer $R_n$ and step sizes $\{\epsilon_{n,t}\}_t$ may not be unique, and in Proposition 4.1, we simply choose the most straightforward feasible candidate. Nonetheless, by this lemma, the nature takes constant step sizes $1 - a$, which implies that we have an inflexible (virtual) firm whose learning rate is always very fast.

By viewing reference prices as prices posted by nature, the induced 3-firm game is also associated with the static game that involves 3 players $i = 1, 2, n$ with respective payoffs $\{\widetilde{\pi}_i\}_{i=1,2,n}$ and common action set $\mathcal{P}$. It turns out that the *pure strategy Nash Equilibrium (PSNE)* of this static game is unique and is identical to the SNE of Lemma 3.2:

**Proposition 4.2** (PSNE of induced 3-firm static game). *Consider the static game with players $i = 1, 2$ and nature $n$, who aims to minimize respective costs $\widetilde{\pi}_1, \widetilde{\pi}_2, \widetilde{\pi}_n$ with identical action set $\mathcal{P} = [\underline{p}, \bar{p}]$. Then, under Assumption 1, this game admits a unique PSNE $(\boldsymbol{p}^*, r^*)$, i.e., $\widetilde{\pi}_i(p_i^*, \boldsymbol{p}_{-i}) \geq \widetilde{\pi}_i(p_i, \boldsymbol{p}_{-i}^*)$ for $\forall p_i \in \mathcal{P}$ and $i = 1, 2, n$. Furthermore, this PSNE is identical to the interior SNE of Lemma 3.2.*

## 5 Convergence Results

The key challenge in showing convergence for the induced 3-firm OMD game play in Algorithm 2 lies in the fact that the step size sequence for nature is the constant $1 - a$, unlike previously studied multi-agent learning settings where step size sequences are typically identical across agents (see for example [44, 38, 10, 46, 35]). This highlights the fundamental issue in our problem of interest: *will convergence still occur if one of the players takes a constant (fixed) step size?* In Section 5.1, we show that prices and reference prices converge to the unique interior SNE when the two firms adopt decreasing step sizes and characterize the corresponding convergence rate. In Section 5.2, we show that there exist constant step sizes for the two firms with which prices convergence to the SNE at faster rates compared to decreasing step sizes.

### 5.1 Decreasing Step Sizes

The first key result in this section is the following theorem, which states that if the two firms run the OMD algorithm with decreasing step sizes that do not go to zero too fast, then convergence to the SNE is guarantied.

**Theorem 5.1** (Convergence under Decreasing Step Sizes). *Suppose that Assumption 1 holds and firm $i = 1, 2$ adopts regularizer $R_i$ that is $\sigma_i$-strongly convex, continuously differentiable, and satisfies the reciprocity condition (see Section 4). Then, when the sequence $\{\epsilon_{i,t} = \epsilon_t\}_t$ is nonincreasing with $\lim_{t\to\infty} \epsilon_t = 0$, we have $\lim_{t\to\infty} (\sum_{i\in[2]} \theta_i p_{i,t} - r_t) \to 0$. Furthermore, if $\lim_{T\to\infty} \sum_{t=1}^{T} \epsilon_t = \infty$, $\lim_{T\to\infty} \sum_{t=1}^{T} \epsilon_t^2 < \infty$ and the sensitivity margin $m \geq 1$, then $\{\boldsymbol{p}_t, r_t\}_t$ converges to the unique interior SNE $(\boldsymbol{p}^*, r^*)$.*

The first part of Theorem 5.1 shows that prices stabilize when the firms' step sizes go to zero eventually. This is an interesting result because in the induced 3-firm dynamic game presented in Algorithm 2, nature adopts a constant step size and learns quickly, while the two other firms are learning slowly through decreasing step sizes. However, firms' prices may not necessarily converge,

and even if they do, firms may have the incentive to deviate, leading to an volatile market.[10] The second part of the theorem addresses this concern and shows that when $\lim_{T\to\infty}\sum_{t=1}^{T}\epsilon_{i,t}=\infty$ and $\lim_{T\to\infty}\sum_{t=1}^{T}\epsilon_{i,t}^2<\infty$, the market becomes stable as the prices converge to the SNE. In fact, these conditions admit a large range of step sizes, e.g. $\epsilon_{i,t}=\Theta(1/t^\eta)$ for $\eta\in(\frac{1}{2},1]$. The proof is provided in Appendix D. Here, we provide some examples to solidify the aforementioned ideas.

**Example 1** (Decreasing Step Sizes). Consider the following demand and reference update model parameters: $\boldsymbol{\alpha}=(5,6)$, $\boldsymbol{\beta}=(2,3)$, $\boldsymbol{\delta}=(0.4,0.7)$, $\boldsymbol{\gamma}=(0.1,0.5)$, $\theta_1=0.8$, $a=0.4$, $\mathcal{P}=[1,2]$, and initial prices $(\boldsymbol{p}_1,r_1)=(1,1,1.5)$. These parameters admit the unique SNE given by $(\boldsymbol{p}^*,r^*)=(1.41,1.28,1.39)$. We consider two different decreasing step size sequences when both firms use the quadratic regularizer, i.e. $R_1(p)=R_2(p)=p^2/2$:

- With $\epsilon_{i,t}=0.1/t^2$, the price profile eventually converges to the point $(\widetilde{\boldsymbol{p}},\widetilde{r})=(1.21,1.18,1.20)$ which is not the SNE (see Figure 1a) and firms are incentivized to deviate, e.g., the best response for firm 1 w.r.t. $\widetilde{p}_2=1.18$ and $\widetilde{r}=1.20$ is $1.40\neq\widetilde{p}_1$.[11] Hence, under this step size sequence, firms may go through different epochs in the long run, in which firms converge in an epoch, and may decide to deviate and start over.

- With $\epsilon_{i,t}=1/t$, we have $\lim_{T\to\infty}\sum_{t=1}^{T}\epsilon_{i,t}=\infty$ and $\lim_{T\to\infty}\sum_{t=1}^{T}\epsilon_{i,t}^2<\infty$. Thus, per Theorem 5.1, prices and reference prices converge to the unique SNE; see Figure 1b. Moreover, we observe that (i) convergence occurs very quickly (for $t\geq 20$), and (ii) prices do not converge monotonically. The latter is in contrast with the pricing policy presented in Theorem 3.1.

In Example 1, we observe fast convergence to the SNE when firms choose decreasing step sizes. Inspired by this, we also characterize convergence rates for such step sizes:

**Theorem 5.2** (Convergence Rate under Decreasing Step Sizes). *Assume Assumption 1 holds. For any sensitivity margin $m\geq 2$, if both firms adopt regularizer $R_i(z)=z^2$, there exists step sizes $\epsilon_{i,t}=\Theta(1/t)$ and an absolute constant $c$, which depends on $a$ and $\max\{\theta_1,\theta_2\}$, such that $\|\boldsymbol{p}^*-\boldsymbol{p}_t\|^2\leq c/t$ for any $t\in\mathbb{N}^+$.*

The proof of this theorem constructs a sufficiently large absolute constant $c$ and shows $\|\boldsymbol{p}^*-\boldsymbol{p}_t\|^2\leq c/t$ via induction. The main procedure involves bounding $\|\boldsymbol{p}^*-\boldsymbol{p}_{t+1}\|^2$ with $\|\boldsymbol{p}^*-\boldsymbol{p}_t\|^2$ and $|r_t-r^*|$, and developing a tight bound for $\sum_{\tau=1}^{t-1}\|\boldsymbol{p}^*-\boldsymbol{p}_\tau\|^2$. Bounding $\sum_{\tau=1}^{t-1}\|\boldsymbol{p}^*-\boldsymbol{p}_\tau\|^2$ helps us bound $|r_t-r^*|$ because the deviations of prices w.r.t. the interior SNE will cumulatively propagate into $|r_t-r^*|$ due to reference price update dynamics. The detailed proof is provided in Appendix D. We also remark that the condition $m\geq 2$ is a rather practical regime because this condition, as discussed in Section 2, implies a firm's demand is more sensitive to its own prices compared to competitor's prices and surcharge (or discounts) relative to reference prices. Finally, we remark that the constant $c$ scales reasonably w.r.t. $a$ and $\max\{\theta_1,\theta_2\}$ as long as they are bounded away from 1; see Figure 2b in Appendix D.7 for an illustration for $a\in[0.1,0.9]$ and $\max\{\theta_1,\theta_2\}\in\{0.5,0.6,\ldots,0.9\}$.

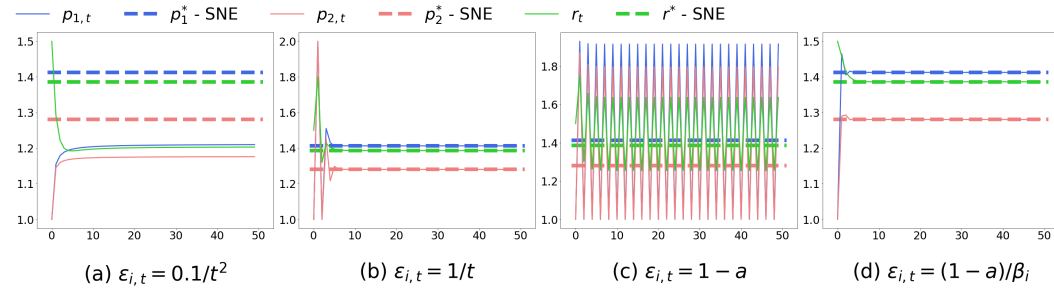

Figure 1: Illustration of price and reference price trajectories in Examples 1 and 2 under different step size sequences. The y-axis represents price levels, as the x-axis denotes time.

## 5.2 Constant Step Sizes

We start by revisiting Example 1 and adopt constant step sizes.

**Example 2** (Constant Step Sizes). Consider the same demand and reference update model parameters in Example 1.

- With $\epsilon_{i,t} = 1 - a$, Figure 1c shows price profiles do not converge and oscillate in the long-run.

- With $\epsilon_{i,t} = (1-a)/\beta_i$, Figure 1d shows price profiles converge to the SNE at a faster rate compared to decreasing step sizes in Figure 1b.

Given this example, we present the first main result for this section in the following theorem (see proof in Appendix D) which shows that under some conditions, there exists constant step size proportional to $\frac{1-a}{\beta_i}$ under which pricing profiles and reference price convergence to the unique interior SNE.

**Theorem 5.3** (Sufficient Conditions for Convergence under Constant Step Sizes). *Suppose that firm $i$ adopts regularizer $R_i$ that is $\sigma_i$-strongly convex and continuously differentiable. For strong-convexity parameters $\sigma_1, \sigma_2$ and sensitivity margin $m$, define the set $\mathcal{S}_{i,m} = \{z > 0 : f_{i,m}(z) < 0\}$, where*

$$f_{i,m}(z) = \begin{cases} \left(4\sigma_i + \frac{2\sigma_{-i}}{m^2}\right) z^2 - \left(\left(2 - \frac{1}{2m}\right)\sigma_i - \frac{\sigma_{-i}}{2m}\right) z + \frac{3}{4} & i = 1, 2 \\ \frac{2}{m^2}\left(\sigma_1 + \sigma_2\right) z^2 + \frac{1}{2m}\left(\sigma_1 + \sigma_2\right) z - \frac{1}{4} & i = n \end{cases}. \tag{6}$$

*Then, under Assumption 1, if $\cap_{i=1,2,n}\mathcal{S}_{i,m} \neq \emptyset$, the step size sequence $\epsilon_{i,t} = s\sigma_i\frac{(1-a)}{\beta_i}$ ($i = 1, 2$) for any $s \in \cap_{i=1,2,n}\mathcal{S}_{i,m}$ guarantees $\{\boldsymbol{p}_t, r_t\}_t$ converges to the unique interior SNE $(\boldsymbol{p}^*, r^*)$.*

This theorem indicates that under some conditions on $m, \sigma_1$, and $\sigma_2$, there exist constant step sizes with which convergence to the unique interior SNE is guaranteed. The desired step size is proportional to $\frac{1-a}{\beta_i}$. This, roughly speaking, implies that prices converge to the SNE if firms adjust prices at a pace similar to that of nature. Recall that $1 - a$ can be considered as the step size of nature, and by demand model in Equation (1), $\beta_i$ is firm $i$'s price sensitivity parameter. The conditions on $m, \sigma_1$, and $\sigma_2$ in Theorem 5.3 are, in fact, quite mild: the following Corollary 5.3.1 provides an example where for any $m > 2$, we can find sufficiently large $\sigma_1 = \sigma_2$ that guaranties convergence to an SNE.

**Corollary 5.3.1** (Convergence under Constant Step Sizes). *For any sensitivity margin $m > 2$, assume both firms adopt continuously differentiable regularizer $R_i$ that is $\sigma$-strongly convex where $\sigma > \sigma_0$ and $\sigma_0 := \max\left\{\frac{6(2m^2+1)}{(2m-1)^2}, \frac{(2m^2+7)^2}{8m^3-36m+8}\right\}$. Then there exists constant $s$ dependent on $m$ and $\sigma$ so if firm $i \in \{1, 2\}$ adopts step size $\epsilon_{i,t} = s\sigma\frac{(1-a)}{\beta_i}$, $\{\boldsymbol{p}_t, r_t\}_t$ converges to the unique interior SNE.*

This corollary provides sufficient conditions for the existence of constant step sizes that guarantee convergence to the SNE for any sensitivity margin $m > 2$. In fact, for suitable $m$, we can possibly find relatively small values of $\sigma_1$ and $\sigma_2$ such that the conditions are satisfied (e.g., $\sigma_1 = \sigma_2 = 4$ for $m = 5$). Note that $\sigma_0 = \Theta(m)$ for large $m$, which means firms generally need to take larger strong-convexity parameters as $m$ increases. (See Figure 2a in Appendix D.7 for illustration of $\sigma_0$ as a function of $m$.) Having everything else fixed, the larger $\sigma$, the slower price movements happens. [12] This is so because for large $m$, a firm's demand is very sensitive to its own prices, encouraging the firm to adjust prices slowly via large $\sigma$.

Moreover, we also characterize the convergence rate when firms adopt suitable constant step sizes via the following Theorem 5.4, and highlight that such fast learning rates give us much faster convergence to the SNE, compared to slow learning rates from decreasing step sizes.

**Theorem 5.4** (Convergence Rate for Constant Step Sizes). *For any sensitivity margin $m > 2$, assume that both firms use quadratic regularizer $R_i(z) = \frac{\sigma z^2}{2}$ for any $\sigma > \sigma_0$, where $\sigma_0$ is defined in Corollary 5.3.1. Then, under Assumption 1 there exists constant $s > 0$, dependent on $m$ and $\sigma$, such that if firm $i = 1, 2$ adopts step size $\epsilon_{i,t} = s\sigma\frac{(1-a)}{\beta_i}$ for $t \in \mathbb{N}^+$, we have $\|\boldsymbol{p}^* - \boldsymbol{p}_t\|^2 \leq \frac{1+2\sigma}{\sigma}\left(\bar{p} - \underline{p}\right)^2 \left(\frac{1+a}{2}\right)^t$.*

## 5.3 Comparison with Multi-agent Online Learning

In light of Proposition 4.2, we can characterize the 3-player game consisting of firms and nature with the mapping $\boldsymbol{g} : \mathbb{R}_+^3 \to \mathbb{R}_+^3$ s.t. $\boldsymbol{g}(\boldsymbol{p}) = (\partial\widetilde{\pi}_i/\partial p_i)_{i=1,2,n}$, where we slightly abuse the notation and

write $\boldsymbol{p} = (p_1, p_2, p_n)$, and $\boldsymbol{g}(\boldsymbol{p}) = (g_1(\boldsymbol{p}), g_2(\boldsymbol{p}), g_n(\boldsymbol{p}))$. Note that the corresponding Jacobian of $\boldsymbol{g}$ is

$$J = \begin{pmatrix} 2\beta_1, -\delta_1, -\gamma_1 \\ -\delta_2, 2\beta_2, -\gamma_2 \\ -\theta_1, -\theta_2, 1 \end{pmatrix},$$

which is not necessarily positive definite,[13] despite being diagonally dominant[14] due to our assumptions on model parameters as illustrated in Section 2. Also note that $\boldsymbol{g}(\boldsymbol{p}) = J\boldsymbol{p}$ is linear in $\boldsymbol{p}$, and Corollary 1.4 in [37] implies $\boldsymbol{g}$ is monotone if and only if $J$ is positive definite. Hence, in our setting, the mapping $\boldsymbol{g}$ may not be monotone, which prohibits us from naively applying arguments in the variational inequality (VI) framework to conclude convergence of the system as agents run OMD (see [44, 38] for a detailed introduction on convergence to Nash Equilibrium under the VI framework).

Consequently, our proof techniques for Theorems 5.1, 5.2, 5.3, and 5.4 are not standard since the aforementioned mapping $\boldsymbol{g}$ does not necessarily satisfy monotonicity or other favorable properties that allow direct applications of the VI methodology. Even if we assume $\boldsymbol{g}$ is monotonic, we still face technical issues that arise from heterogeneous step sizes, which provides another motivation to develop new techniques to show convergence as firms run general OMD algorithms. To briefly illustrate such challenges, assume $\boldsymbol{g}$ is monotonic, meaning $\langle \boldsymbol{g}(\boldsymbol{p}), \boldsymbol{p}^* - \boldsymbol{p} \rangle \le \langle \boldsymbol{g}(\boldsymbol{p}^\star), \boldsymbol{p}^* - \boldsymbol{p} \rangle = 0$ for $\forall \boldsymbol{p} \in \mathcal{P}^3$, where the equality follows from Assumption 1 and first order conditions. If one can enforce $\epsilon_{i,t} = \epsilon_t$ for $i = 1, 2, n$, showing the convergence results in this line of work boils down to verifying the following inequalities (e.g. see [10, 46, 35]):

$$\sum_{i=1,2,n} D_i(p_i^*, p_{i,t+1}) \overset{(a)}{\le} \sum_{i=1,2,n} D_i(p_i^*, p_{i,t}) + \epsilon_t \langle \boldsymbol{g}(\boldsymbol{p}_t), \boldsymbol{p}^* - \boldsymbol{p}_t \rangle + \epsilon_t^2 c_2 \overset{(b)}{<} \sum_{i=1,2,n} D_i(p_i^*, p_{i,t}).$$

where $c_2$ can be viewed as some absolute constant, and $D_i$ is Bregman divergence w.r.t. strongly convex regularizer $R_i$. At a high level, the above equations show that the distance between $p_i^*$ and $p_{i,t}$ becomes smaller over time and hence implies convergence to the SNE. The inequality (a) follows from classical mirror descent proofs; and inequality (b) utilizes the variational stability condition by choosing suitable $\epsilon_t$ (for example $\epsilon_t = \Theta(1/t)$). However, this procedure will not be applicable in our setting as nature is inflexible in the sense that it always takes the constant step size sequence $1 - a$, while the two firms are unaware of how nature updates, and may independently use different step sizes (e.g. decreasing step sizes).

## Footnotes

[1]We consider a linear demand model, but firms are not aware of the functional form of the demand.

[2]OMD algorithms are closely related to the regularized learning paradigm, which includes algorithms such as follow the regularized leader (FTRL), EXP3, Hedge, etc (see [26] for a comprehensive survey).

[3]The dependency of demand on price surcharges and discounts are of the same order $\gamma_i$, which corresponds to so-called risk-neutral consumers. Related literature have also studied asymmetric demand dependencies on surcharges and discounts; see [42, 39, 28].

[4]We note that such information can be obtained by a slight perturbation of the posted price. Furthermore, the assumption of having access to the first-order oracle is very common in the literature; see, for example, a comprehensive introduction to convex optimization in [41].

[5]Here, the revenue function $\pi_i$ is quadratic, so $\operatorname{argmax}_{p \in \mathcal{P}} \pi_i$ is a singleton.

[6]A set $\mathcal{A} \subset \mathbb{R}^d$ is an ordered set with total ordering if for any $\boldsymbol{x}, \boldsymbol{y} \in \mathcal{A}$, either $\boldsymbol{x} \leq \boldsymbol{y}$ or $\boldsymbol{y} \leq \boldsymbol{x}$ where the relationship $\leq$ and $\geq$ between two vectors is component-wise.

[7]The Bregman divergence $D : \mathcal{C} \times \mathcal{C} \to \mathbb{R}^+$ associated with convex and continuously differentiable regularizer function $R : \mathcal{C} \to \mathbb{R}$ and convex set $\mathcal{C} \subset \mathbb{R}$ is defined as $D(x, y) := R(x) - R(y) - R'(y)(x - y)$.

[8]Firms do not know the linear form of demand, and hence cannot learn parameters and then best respond given parameter estimates.

[9]$y_{i,t+1}$ exists when $R_i$ is continuously differentiable and convex, see Section 3.3 of [9] or Section 5.2 of [12]

[10]An example is the extreme case where firms 1 and 2 adopt step sizes $\epsilon_{i,t}=0$. This obviously guaranties convergence because prices are fixed at the initial prices, which are likely not the SNE, encouraging the firms to unilaterally deviate.

[11]Here, we choose $\epsilon_{i,t}=0.1/t^2$ because the gap between the convergence point and the SNE is more visible. For the more natural choice $\epsilon_{i,t}=1/t^2$, we obtain similar results.

[12]For example, taking $R_i(z) = \sigma z^2$ in step 5 of Algorithm 1, we get $y_{i,t+1} = p_{i,t} - \frac{\epsilon_{i,t}g_{i,t}}{\sigma}$, which implies the gap between $y_{i,t+1}$ and $p_{i,t}$ is small with large $\sigma$.

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
