[Supplementary Material]

Appendices for

# No-regret Learning in Price Competitions under Consumer Reference Effects

## A Expanded Literature Review

**Reference Price Effects and Monopolist Pricing.** Consumer reference effects have been validated empirically in many works including [47, 48, 30, 7]. This motivated a wide range of research including [32, 22, 42, 3, 39] that studies optimal dynamic monopolistic pricing under different demand and reference price update models, where the single firm has complete information on consumer demand as well as how reference prices update. There are also very recent works that address the dynamic pricing problem with consumer reference effects under uncertain demand. [7] utilizes real retail data and concludes the inclusion of exposure effects to sales or number of consumers[15] when considering reference price formations leads to more accurate forecasts in demand, and proposes a pricing policy using dynamic programming. [17] couples the problem of monopolistic dynamic pricing with reference effects and online demand learning. In our work, similar to [7, 17], firms do not know the demand functions and how the reference prices are formed. But, while in [7, 17], the form of demand model is known to the firm (monopolist) that aims to estimate model parameters, our work assumes competition between firms that do not know the form of demand and hence run OMD algorithms to set their prices. Additionally, the algorithms proposed in [7] and [17] aim to increase revenue from the firm's perspective, while our work focuses on analyzing market stability for long-run competitions under reference effects.

**Pricing in Competitive Markets without Reference Effects.** A large stream of work studies static price competitions and characterizes structural properties of corresponding equilibria (for example, see [8, 24, 4]). Other works such as [1, 33, 23] study oligopolistic dynamic pricing under various inventory, market, or product characteristics. Nevertheless, these two lines of works are oblivious to consumer reference effects. In this work, we jointly tackle the dynamic pricing problems in competitive markets with reference price effects when the firms lack the knowledge of demand functions and reference price dynamics.

**Pricing in Competitive Markets with Reference Effects.** Similar to our work, the works of [16] and [21] also consider price competitions under reference effects. [16] considers a similar linear demand model and an identical reference price update dynamic, but the work only provides theoretical analysis on the two-firm, two-period price competition setting, for which they characterize the unique sub-game perfect Nash Equilibrium. On the other hand, [21] studies multiple-firm single-period price competition equipped with different reference price effects in consumers' demand (e.g. the reference price is specified by the lowest posted price). Additionally, both of these works study the complete information setting. In contrast to these two papers, our work studies price competitions over an infinite time horizon where reference prices adjust over time, and provides theoretical guarantees for the convergence of pricing strategies under the partial information setting. Finally, our work is the first study that provides theoretical analyses on long-term market stability of repeated price competitions in the presence of consumer reference effects.

**Convergence in Games with Descent Methods.** In addition to [34, 10, 35] that we discussed in Section 1, here we also review related literature that study convergence in games where multiple agents adopt descent methods. [43] studies finding a Nash Equilibrium of concave games via having each agent run projected gradient descent under complete information, i.e., agents know each others' payoff functions and decision constraints. [40] studies a distributed network optimization problem to optimize a sum of convex objective functions corresponding to multiple agents. Our paper distinguishes itself from this line of work from two aspects: unlike the two aforementioned works, (i) our model involves a varying underlying state (i.e., reference prices) dependent on all agents' historical decisions, and can be modeled as a sequence of decisions made by an inflexible virtual agent that adopts descent methods with a constant step size; (ii) the agents (i.e., firms) in our model do

not have any information on one another's revenue function or how reference prices update. Finally, [6] considers multiple budget-constrained bidders participating in repeated second price auctions by adopting so-called *adaptive pacing strategies*, which is equivalent to the subgradient descent method. In their setting, the subgradient for each bidder's objective is a function of all bidders' decisions as well as its budget rate (i.e. total fixed budget divided by a given time horizon), which can be thought of as an underlying model state that remains constant over time.[16] In contrast, in our setting, the gradient oracle each firm receives is not only a function of all firms' decisions, but also of the reference price which varies over time according to firms' past decisions, making our analysis more challenging.

# B    Appendix for Section 3

**Additional Definitions.**    We define the best-response mapping as $\psi : \mathcal{P}^3 \to \mathcal{P}^2$ such that $\boldsymbol{\psi}(\boldsymbol{p}, r) = (\psi_1(p_2, r), \psi_2(p_1, r))$. Then, we can rewrite the set of best-response profiles w.r.t. reference price $r$, defined in Equation (4), as $\mathcal{B}(r) = \{\boldsymbol{p} \in \mathcal{P}^2 : \boldsymbol{p} = \boldsymbol{\psi}(\boldsymbol{p}, r)\}$. Note that for any SNE $(\boldsymbol{p}^*, r^*)$, we must have $\boldsymbol{p}^* \in \mathcal{B}(r^*)$, and $\boldsymbol{p}^*$ is a fixed point of the mapping $\boldsymbol{\psi}(\cdot, r^*)$.

## B.1    Proof of Theorem 3.1

(i) By first order conditions, we know that

$$\arg\max_{p \in \mathbb{R}} \pi_i(p, p_{-i}, r) = \frac{\alpha_i + \delta_i p_{-i} + \gamma_i r}{2\beta_i} \,.$$

Hence, due to boundary constraints on the decision set $\mathcal{P}$ and the revenue function being quadratic, we have

$$\psi_i(p_{-i}, r) = \arg\max_{p \in \mathcal{P}} \pi_i(p, p_{-i}, r) = \Pi_{\mathcal{P}}\left(\frac{\alpha_i + \delta_i p_{-i} + \gamma_i r}{2\beta_i}\right) \,,$$

where $\Pi_{\mathcal{P}} : \mathbb{R} \to \mathcal{P}$ is the projection operator such that $\Pi_{\mathcal{P}}(z) = z\mathbb{I}\{z \in \mathcal{P}\} + \underline{p}\mathbb{I}\{z < \underline{p}\} + \bar{p}\mathbb{I}\{z > \bar{p}\}$. Hence, $\psi_i(p_{-i}, r)$ is a nondecreasing function in $p_{-i}$ and $r$, which further implies $\boldsymbol{\psi}(\boldsymbol{p}, r)$ is nondecreasing in $\boldsymbol{p}$ and $r$. Again, recall for any $\boldsymbol{x}, \boldsymbol{y}$, the relationships $\boldsymbol{x} \leq \boldsymbol{y}$ and $\boldsymbol{y} \leq \boldsymbol{x}$ are component-wise comparisons.

We now follow a similar proof to that of Tarski's fixed point theorem: consider the set $\mathcal{B}_+(r) = \{\boldsymbol{p} \in \mathcal{P}^2 : \boldsymbol{p} \leq \boldsymbol{\psi}(\boldsymbol{p}, r)\}$. It is apparent that this set is nonempty because $(\underline{p}, \underline{p}) \in \mathcal{B}_+(r)$. Fix any $\boldsymbol{p} \in \mathcal{B}_+(r)$. Then, we have $\boldsymbol{p} \leq \boldsymbol{\psi}(\boldsymbol{p}, r)$ which further implies $\boldsymbol{\psi}(\boldsymbol{p}, r) \leq \boldsymbol{\psi}(\boldsymbol{\psi}(\boldsymbol{p}, r), r)$ since $\boldsymbol{\psi}(\boldsymbol{p}, r)$ is nondecreasing in $\boldsymbol{p}$. Hence $\boldsymbol{\psi}(\boldsymbol{p}, r) \in \mathcal{B}_+(r)$. By taking $\boldsymbol{U}(r) = \sup \mathcal{B}_+(r)$ (this is possible since all $\boldsymbol{p} \in \mathcal{B}_+(r)$ are bounded), we have $\boldsymbol{p} \leq \boldsymbol{U}(r)$ so $\boldsymbol{p} \leq \boldsymbol{\psi}(\boldsymbol{p}, r) \leq \boldsymbol{\psi}(\boldsymbol{U}(r), r)$. This further implies $\boldsymbol{U}(r) \leq \boldsymbol{\psi}(\boldsymbol{U}(r), r)$ because $\boldsymbol{U}(r)$ is the least upper bound of $\mathcal{B}_+(r)$, and thus $\boldsymbol{U}(r) \in \mathcal{B}_+(r)$. This allows us to conclude $\boldsymbol{\psi}(\boldsymbol{U}(r), r) \leq \boldsymbol{U}(r)$ and hence $\boldsymbol{U}(r) = \boldsymbol{\psi}(\boldsymbol{U}(r), r)$, which means $\boldsymbol{U}(r) = \sup \mathcal{B}_+(r)$ is a fixed point of the mapping $\boldsymbol{\psi}(\cdot, r)$. Thus, $\boldsymbol{U}(r)$ belongs in the set of best-response profiles $\mathcal{B}(r)$, confirming $\mathcal{B}(r)$ is not empty.

Next, we show that $\mathcal{B}(r)$ is an ordered set with total ordering if it is not a singleton. To do so, consider any $\boldsymbol{p}, \boldsymbol{q} \in \mathcal{B}(r)$ and without loss of generality assume $p_1 > q_1$. Since $p_1 = \psi_1(p_2, r)$ and $q_1 = \psi_1(q_2, r)$, by monotonicity of $\psi_1(\cdot, r)$ we have $p_2 > q_2$. Thus, $\boldsymbol{p} > \boldsymbol{q}$ and $\mathcal{B}(r)$ is an ordered set with total ordering.

(ii) In the proof of (i), we showed that $\boldsymbol{U}(r) = \sup \{\boldsymbol{p} \in \mathcal{P}^2 : \boldsymbol{p} \leq \boldsymbol{\psi}(\boldsymbol{p}, r)\}$ is a fixed point of the best-response mapping $\boldsymbol{\psi}(\cdot, r)$ for any $r$ which allows us to conclude $\boldsymbol{U}(r)$ is the largest best-response profile, i.e., $\boldsymbol{U}(r) = \max \mathcal{B}(r)$, and hence $\boldsymbol{p}_t = \boldsymbol{U}(r_t)$. Furthermore, since $\boldsymbol{\psi}(\boldsymbol{p}, r)$ is increasing in $r$, we know that $\boldsymbol{U}(\cdot) = \sup \{\boldsymbol{p} \in \mathcal{P}^2 : \boldsymbol{p} \leq \boldsymbol{\psi}(\boldsymbol{p}, \cdot)\}$ is also an increasing function. In the following, we will argue that the reference prices $r_t$ is monotonically increasing or decreasing, which implies $\boldsymbol{p}_t = \boldsymbol{U}(r_t)$ is also monotonic, and hence converges since prices and reference prices are bounded.

We write $\boldsymbol{U}(r) = (U_1(r), U_2(r))$. At $t = 1$, if $\theta_1 p_{1,1} + \theta_2 p_{2,1} = \theta_1 U_1(r_1) + \theta_2 U_2(r_1) \geq r_1$, then the reference price at $t = 2$ satisfies the following equation

$$r_2 = a r_1 + (1 - a)(\theta_1 p_{1,1} + \theta_2 p_{2,1}) \geq r_1 \,.$$

By the monotonicity of $\boldsymbol{U}(\cdot)$, we have $p_{i,2} = U_i(r_2) \geq U_i(r_1) = p_{i,1}$ for $i = 1, 2$. Thus,

$$
\begin{aligned}
r_3 &= ar_2 + (1-a)\left(\theta_1 p_{1,2} + \theta_2 p_{2,2}\right) \\
&\geq ar_1 + (1-a)\left(\theta_1 p_{1,1} + \theta_2 p_{2,1}\right) \\
&= r_2 \,.
\end{aligned}
$$

A simple induction argument thus shows $\{r_t\}_t$ is a nondecreasing sequence. Since $r_t \leq \bar{p}$ for any $t \in \mathbb{N}$, we know that $\{r_t\}_t$ converges to some number $r_+ \in [\underline{p}, \bar{p}]$ when $\theta_1 p_{1,1} + \theta_2 p_{2,1} \geq r_1$. Furthermore, we observe that $\lim_{t\to\infty} \boldsymbol{\psi}(\boldsymbol{U}(r_t), r_t) = \boldsymbol{\psi}(\boldsymbol{U}(r_+), r_+)$ by the definition of $\boldsymbol{\psi}$. Also, from (i) we have $\boldsymbol{\psi}(\boldsymbol{U}(r_t), r_t) = \boldsymbol{U}(r_t)$ and $\boldsymbol{\psi}(\boldsymbol{U}(r_+), r_+) = \boldsymbol{U}(r_+)$ because $\boldsymbol{U}(r)$ is a fixed point of $\boldsymbol{\psi}(\cdot, r)$ for any $r$. Hence, $\lim_{t\to\infty} \boldsymbol{U}(r_t) = \boldsymbol{U}(r_+)$, which implies $\{\boldsymbol{p}_t = \boldsymbol{U}(r_t)\}_t$ converges to $\boldsymbol{U}(r_+)$. Note that convergence is monotonic because $\boldsymbol{U}(\cdot)$ is nondecreasing. Therefore,

$$
\theta_1 U_1(r_+) + \theta_2 U_2(r_+) = \lim_{t\to\infty} \theta_1 U_1(r_t) + \theta_2 U_2(r_t) = \lim_{t\to\infty} r_{t+1} = r_+ \,,
$$

which implies $(\boldsymbol{U}(r_+), r_+)$ is an SNE. We can thus conclude that if $\theta_1 p_{1,1} + \theta_2 p_{2,1} = \theta_1 U_1(r_1) + \theta_2 U_2(r_1) \geq r_1$, firms' prices and reference prices converge monotonically to an SNE $(\boldsymbol{U}(r_+), r_+)$.

Following a symmetric argument, if $\theta_1 p_{1,1} + \theta_2 p_{2,1} < r_1$, we can show that $\{r_t\}_t$ is a nonincreasing sequence. Since $r_t \geq \underline{p}$ for any $t \in \mathbb{N}$, we know that $\{r_t\}_t$ converges to some number $r_- \in [\underline{p}, \bar{p}]$. Similar to the previous arguments, we can conclude that prices and reference prices converge monotonically to an SNE $(\boldsymbol{U}(r_-), r_-)$.

### B.2 Proof of Lemma 3.2

Let $(\boldsymbol{p}^*, r^*) \in (\underline{p}, \bar{p})^3$ be an interior SNE, whose existence is guarantied by Assumption 1. Since revenue functions are quadratic, first order conditions at the interior best-response profiles should hold, which means the derivative of revenue functions at the interior best-responses $p_1^* = \psi_1(p_2^*, r^*)$ and $p_2^* = \psi_2(p_1^*, r^*)$ should be 0:

$$
\frac{\partial \pi_1(\boldsymbol{p}^*, r^*)}{\partial p_1} = \frac{\partial \pi_2(\boldsymbol{p}^*, r^*)}{\partial p_2} = 0 \,,
$$

which leads to the relationship $\alpha_1 - 2\beta_1 \psi_1(p_2^*, r^*) + \delta_1 p_2^* + \gamma_1 r^* = \alpha_2 - 2\beta_2 \psi_2(p_1^*, r^*) + \delta_2 p_1^* + \gamma_2 r^* = 0$. Solving for the best-response equations, we get

$$
p_1^* = \psi_1(p_2^*, r^*) = \frac{\alpha_1 + \delta_1 p_2^* + \gamma_1 r^*}{2\beta_1} \quad , \quad p_2^* = \psi_2(p_1^*, r^*) = \frac{\alpha_2 + \delta_2 p_1^* + \gamma_2 r^*}{2\beta_2} \,. \tag{7}
$$

Finally, the definition of an SNE guaranties $r^* = \theta_1 p_1^* + \theta_2 p_2^*$. Thus, solving for $(\boldsymbol{p}^*, r^*)$, we obtain the unique solution

$$
\begin{aligned}
p_i^* &= \frac{2\alpha_i \beta_{-i} - \alpha_i \theta_{-i} \gamma_{-i} + \alpha_{-i}\left(\delta_i + \theta_{-i}\gamma_i\right)}{\left(2\beta_1 - \theta_1\gamma_1\right)\left(2\beta_2 - \theta_2\gamma_2\right) - \left(\theta_2\gamma_1 + \delta_1\right)\left(\theta_1\gamma_2 + \delta_2\right)} \quad i = 1, 2 \\
r^* &= \frac{\theta_1\left(2\alpha_1\beta_2 + \alpha_2\delta_1\right) + \theta_2\left(2\alpha_2\beta_1 + \alpha_1\delta_2\right)}{\left(2\beta_1 - \theta_1\gamma_1\right)\left(2\beta_2 - \theta_2\gamma_2\right) - \left(\theta_2\gamma_1 + \delta_1\right)\left(\theta_1\gamma_2 + \delta_2\right)} \,.
\end{aligned} \tag{8}
$$

This implies that under Assumption 1, the interior SNE is unique. We remark that for any $i = 1, 2$, because $\beta_i \geq m(\delta_i + \gamma_i) > 0$ and $m \geq 2 > 1$ we have $2\beta_i - \theta_i\gamma_i > \beta_i - \theta_i\gamma_i > \delta_i + \gamma_i - \theta_i\gamma_i = \theta_{-i}\gamma_i + \delta_i$. Hence, $p_i^*, r^* > 0$.

## C  Appendix for Section 4

### C.1  Proof of Proposition 4.1

First of all, it is easy to see prices at the first period are identical between Algorithm 1 and 2: $p_{i,1} = \arg\max_{p\in\mathcal{P}} R_i$ for $i = 1, 2$ and $p_{n,1} = r_1$. We now use induction to show price trajectories of the two algorithms are identical via considering the induction hypothesis that prices and reference prices are the same up to period $t \in \mathbb{N}^+$.

Note that $R_n(z) = \frac{1}{2}z^2$ implies $R'_n(z) = z$. Then, the proxy variable update step for nature is

$$
\begin{aligned}
y_{n,t+1} &= p_{n,t} - (1-a)\frac{\partial \widetilde{\pi}_n(\boldsymbol{p})}{\partial p_n}\Big|_{\boldsymbol{p}=p_{1,t},p_{2,t},p_{n,t}} \\
&= p_{n,t} - (1-a)\left(p_{n,t} - \theta_1 p_{1,t} - \theta_2 p_{2,t}\right) \\
&= ar_t + (1-a)\left(\theta_1 p_{1,t} + \theta_2 p_{2,t}\right) \\
&= r_{t+1}.
\end{aligned}
$$

Since $y_{n,t+1} = r_{t+1} \in \mathcal{P}$ the projection step for nature is trivial, which means $p_{n,t+1} = y_{n,t+1} = r_{t+1}$. Furthermore, it is not difficult to see that prices $p_{1,t+1} = \Pi_{p\in\mathcal{P}}(y_{1,t+1})$ and $p_{2,t+1} = \Pi_{p\in\mathcal{P}}(y_{2,t+1})$ are identical between the two algorithms under the induction hypothesis. This implies that Algorithm 2 indeed recovers the prices and reference prices produced by Algorithm 1.

### C.2 Proof of Proposition 4.2

Directly considering first order conditions for the cost functions $\{\widetilde{\pi}_i\}_{i=1,2,n}$, we have the system of equations

$$
\begin{aligned}
0 &= \frac{\partial \widetilde{\pi}_1(p_1,p_2,p_n)}{\partial p_1} = 2\beta_1 p_1 - (\alpha_1 + \delta_1 p_2 + \gamma_1 r) \\
0 &= \frac{\partial \widetilde{\pi}_2(p_1,p_2,p_n)}{\partial p_2} = 2\beta_2 p_2 - (\alpha_2 + \delta_2 p_1 + \gamma_2 r) \\
0 &= \frac{\partial \widetilde{\pi}_n(p_1,p_2,p_n)}{\partial p_n} = p_n - (\theta_1 p_1 + \theta_2 p_2) \ .
\end{aligned}
$$

Solving these equations results in a unique solution that is identical to that in Equation (8), which is the unique interior SNE according to Lemma 3.2. Since the SNE is an interior point of $(\underline{p}, \bar{p})^3$, it is the unique PSNE of the induced static 3-firm game.

## D Appendix for Section 5

### D.1 Additional Definitions

**Definition D.1** (Bregman Divergence). The Bregman divergence $D : \mathcal{C} \times \mathcal{C} \to \mathbb{R}^+$ associated with convex set $\mathcal{C} \subset \mathbb{R}$, and convex and continuously differentiable function $R : \mathcal{C} \to \mathbb{R}$ is defined as

$$
D(x,y) := R(x) - R(y) - R'(y)(x-y) \geq 0 \, ,
$$

where the inequality follows from convexity of $R$. Furthermore, if $R$ is $\sigma$-strongly convex, then $D(x,y) \geq \frac{\sigma^2}{2}(x-y)^2$.

Note that $D_i$ is the Bregman divergence associated with regularizer $R_i$ used by firm $i = 1, 2$, and $D_n$ is Bregman divergence associated with regularizer $R_n$ used by nature.

**Definition D.2.** Let $g_i^*$ be the partial derivative of the cost function $\widetilde{\pi}_i$ w.r.t. $p_i$ evaluated at the interior SNE $(\boldsymbol{p}^*, r^*)$, i.e. for $i = 1, 2, n$

$$
g_i^* = \frac{\partial \widetilde{\pi}_i(p_1,p_2,p_n)}{\partial p_i}\Big|_{p_1=p_1^*,p_2=p_2^*,p_n=r^*} \, .
$$

### D.2 Proof for Theorem 5.1

The proof of this theorem is divided into two parts. In the first part, we show that the price profiles $(\boldsymbol{p}_t, r_t)$ converge as $t \to \infty$ under the condition $\lim_{t\to\infty} \epsilon_{i,t} = 0$, $i \in \{1, 2\}$. In the second part, under the additional conditions $\lim_{T\to\infty}\sum_{t=1}^{T}\epsilon_{i,t} = \infty$ and $\lim_{T\to\infty}\sum_{t=1}^{T}\epsilon_{i,t}^2 < \infty$, we show that the price profiles converge to the unique interior SNE.

**First part: Convergence of prices and reference prices.** Recall that $g_{i,t} = g_i(p_{i,t}, r_t) = 2\beta_i p_{i,t} - (\alpha_i + \delta_i p_{-i,t} + \gamma_i r_t)$, and $p_{i,t}$ and $p_{-i,t}$ are both bounded. Hence, because for $i = 1, 2$, $\lim_{t\to\infty} \epsilon_{i,t} = 0$ and $\{\epsilon_{i,t}\}_t$ is nonincreasing, we have for any small $\epsilon > 0$ there exist $t_\epsilon \in \mathbb{N}$ such that $|\epsilon_{i,t} g_{i,t}| \leq \frac{\sigma_i^2 \epsilon}{6}$ for all $t \geq t_\epsilon$. Our goal is to show that for $t \geq t_\epsilon$, $|p_{i,t+1} - p_{i,t}|$ is small.

For $t \geq t_\epsilon$,

$$|p_{i,t+1} - p_{i,t}| \leq |p_{i,t+1} - y_{i,t+1}| + |y_{i,t+1} - p_{i,t}| \stackrel{(a)}{\leq} |p_{i,t+1} - y_{i,t+1}| + \frac{\epsilon}{6}. \tag{9}$$

To see why inequality (a) holds recall that $y_{i,t+1}$ is the proxy variable in Step 5 of Algorithm 1 such that $R_i'(y_{i,t+1}) - R_i'(p_{i,t}) = \epsilon_{i,t} g_{i,t}$. Hence,

$$\sigma_i^2 |y_{i,t+1} - p_{i,t}| \leq |R_i'(y_{i,t+1}) - R_i'(p_{i,t})| = |\epsilon_{i,t} g_{i,t}| \leq \frac{\sigma_i^2 \epsilon}{6}, \quad t > t_\epsilon, i = 1, 2,$$

which implies that $|y_{i,t+1} - p_{i,t}| \leq \frac{\epsilon}{6}$, as desired. Here, the first inequality holds because:

$$\sigma_i^2 (y_{i,t+1} - p_{i,t})^2 \stackrel{(a)}{\leq} (R_i'(y_{i,t+1}) - R_i'(p_{i,t})) (y_{i,t+1} - p_{i,t})$$
$$\leq |R_i'(y_{i,t+1}) - R_i'(p_{i,t})| \cdot |y_{i,t+1} - p_{i,t}|,$$

where (a) follows from summing up $R_i(y_{i,t+1}) - R_i(p_{i,t}) \geq R_i'(p_{i,t})(y_{i,t+1} - p_{i,t}) + \frac{\sigma_i^2}{2}(y_{i,t+1} - p_{i,t})^2$ and $R_i(p_{i,t}) - R_i(y_{i,t+1}) \geq R_i'(y_{i,t+1})(p_{i,t} - y_{i,t+1}) + \frac{\sigma_i^2}{2}(y_{i,t+1} - p_{i,t})^2$ due to strong convexity.

By Equation (9), for $t \geq t_\epsilon$,

$$|p_{i,t+1} - p_{i,t}| \leq |p_{i,t+1} - y_{i,t+1}| \left( \mathbb{I}\{y_{i,t+1} < \underline{p}\} + \mathbb{I}\{y_{i,t+1} \in \mathcal{P}\} + \mathbb{I}\{y_{i,t+1} > \bar{p}\} \right) + \frac{\epsilon}{6}$$
$$= |p_{i,t+1} - y_{i,t+1}| \left( \mathbb{I}\{y_{i,t+1} < \underline{p}\} + \mathbb{I}\{y_{i,t+1} > \bar{p}\} \right) + \frac{\epsilon}{6}, \tag{10}$$

where the equality holds because under the event $y_{i,t+1} \in \mathcal{P}$, no projection occurs and hence, $y_{i,t+1} = p_{i,t+1}$. In the first of the proof, we bound the first two terms in the right hand side, i.e., $|p_{i,t+1} - y_{i,t+1}| \mathbb{I}\{y_{i,t+1} < \underline{p}\}$ and $|p_{i,t+1} - y_{i,t+1}| \mathbb{I}\{y_{i,t+1} > \bar{p}\}$.

To bound $|p_{i,t+1} - y_{i,t+1}| \mathbb{I}\{y_{i,t+1} < \underline{p}\}$, similar to Equation (9) we use $|p_{i,t} - y_{i,t+1}| \leq \frac{\epsilon}{6}$ for $t \geq t_\epsilon$ which implies $p_{i,t} - y_{i,t+1} \leq \frac{\epsilon}{6}$. Thus,

$$y_{i,t+1} \geq p_{i,t} - \frac{\epsilon}{6} \stackrel{(a)}{\geq} \underline{p} - \frac{\epsilon}{6}. \tag{11}$$

where (a) holds because $p_{i,t} \geq \underline{p}$ for any $i, t$. On the other hand, under the event $y_{i,t+1} < \underline{p}$, projection occurs and therefore we have $p_{i,t+1} = \underline{p}$.

This yields

$$|p_{i,t+1} - y_{i,t+1}| \mathbb{I}\{y_{i,t+1} < \underline{p}\} = (\underline{p} - y_{i,t+1}) \mathbb{I}\{y_{i,t+1} < \underline{p}\} \stackrel{(a)}{\leq} \left( \underline{p} - \underline{p} + \frac{\epsilon}{6} \right) = \frac{\epsilon}{6}, \tag{12}$$

where (a) follows from Equation (11).

Using a similar argument as above to bound $|p_{i,t+1} - y_{i,t+1}| \mathbb{I}\{y_{i,t+1} > \bar{p}\}$, we have $|p_{i,t+1} - y_{i,t+1}| \mathbb{I}\{y_{i,t+1} > \bar{p}\} \leq \frac{\epsilon}{6}$ under the event $y_{i,t+1} > \bar{p}$.

Hence, plugging these upper bounds back into Equation (9), we can show that for any $\epsilon > 0$ and $t \geq t_\epsilon$

$$|p_{i,t+1} - p_{i,t}| \leq \frac{\epsilon}{6} + \frac{\epsilon}{6} + \frac{\epsilon}{6} = \frac{\epsilon}{2}, \quad i = 1, 2. \tag{13}$$

Now, for any $t \geq t_\epsilon$ we have

$$\left| r_{t+1} - \sum_{i=1,2} \theta_i p_{i,t+1} \right| = \left| ar_t - (1-a) \sum_{i=1,2} \theta_i p_{i,t} - \sum_{i=1,2} \theta_i p_{i,t+1} \right| \tag{14}$$

$$\leq a \left| r_t - \sum_{i=1,2} \theta_i p_{i,t} \right| + \sum_{i=1,2} \theta_i \left| p_{i,t} - p_{i,t+1} \right| \tag{15}$$

$$\leq a \left| r_t - \sum_{i=1,2} \theta_i p_{i,t} \right| + \frac{\epsilon}{2}, \tag{16}$$

where the final inequality follows from Equation (13). Telescoping from $t$ down to $t_\epsilon$, we have

$$\left| r_{t+1} - \sum_{i=1,2} \theta_i p_{i,t+1} \right| \leq a^{t-t_\epsilon+1} \left| r_{t_\epsilon} - \sum_{i=1,2} \theta_i p_{i,t_\epsilon} \right| + \frac{\epsilon}{2} \sum_{\tau=t_\epsilon}^{t} a^{\tau-t_\epsilon}$$

$$\leq a^{t-t_\epsilon+1} \left| r_{t_\epsilon} - \sum_{i=1,2} \theta_i p_{i,t_\epsilon} \right| + \frac{\epsilon}{2(1-a)}$$

$$\leq a^{t-t_\epsilon+1} (\bar{p} - \underline{p}) + \frac{\epsilon}{2(1-a)}.$$

Letting $t \to \infty$ and $\epsilon \to 0$ concludes $\sum_{i=1,2} \theta_i p_{i,t} \to r_t$ for $t \to \infty$.

**Second part: Convergence to the SNE.** The proof of this part is inspired by the proof of Theorem 4.6 in [35]. However, in that proof, they rely on the Nash Equilibria of the game being *variationally stable*, or more strictly speaking, the gradient of the virtual 3-player game (consisting of firms and nature) $g : \mathbb{R}_+^3 \to \mathbb{R}_+^3$ s.t. $g(p) = (\partial \tilde{\pi}_i / \partial p_i)_{i=1,2,n}$ to be a monotone mapping. In this proof, we do not rely on such structural assumption for the virtual 3-player game (see discussion in Section 5.3).

The proof of this theorem is split into two steps. First, we show that for any $\epsilon > 0$, the price profile $(p_{1,t}, p_{2,t})$ must enter an $\epsilon$-neigborhood of the SNE prices $(p_1^*, p_2^*)$ infinitely many times. In the second step, we show that when $(p_{1,t}, p_{2,t})$ enters the $\epsilon$-neigborhood with small enough step sizes, it must remain their forever.

Before we begin, we first introduce a lemma that would be used in both steps:

**Lemma D.1.** *Assume for any $i = 1,2$, $\beta_i \geq m(\delta_1 + \delta_2 + \max\{\gamma_1, \gamma_2\})$ for $m \geq 1$ as described in Section 2. Consider the 2-by-2 matrix $M$*

$$M = \begin{pmatrix} 2\beta_1 - \theta_1 \gamma_1 & -(\delta_1 + \theta_2 \gamma_1) \\ -(\delta_2 + \theta_1 \gamma_2) & 2\beta_2 - \theta_2 \gamma_2 \end{pmatrix}$$

$$M + M^\top = \begin{pmatrix} 4\beta_1 - 2\theta_1 \gamma_1 & -(\delta_1 + \delta_2 + \theta_2 \gamma_1 + \theta_1 \gamma_2) \\ -(\delta_1 + \delta_2 + \theta_2 \gamma_1 + \theta_1 \gamma_2) & 4\beta_2 - 2\theta_2 \gamma_2 \end{pmatrix}, \tag{17}$$

*Then, for any $x \neq 0 \in \mathbb{R}^2$, $x^\top M x > 0$. Furthermore, for any $\epsilon > 0$, define an $\epsilon$-neighborhood around $(p_1, p_2) \in \mathbb{R}^2 : \mathcal{N}_\epsilon(p_1, p_2) = \{x \in \mathbb{R}^2 : \sum_{i=1,2} D_i(x_i, p_i) < \epsilon\}$ where $D_i(x, y) = R(x) - R(y) - R'(y)(x - y)$ is the Bregman divergence w.r.t. $R_i$. Assuming the regularizers satisfy the reciprocity condition, i.e. whenever $x \to y$ for $x, y \in \mathbb{R}$ we have $D_i(x, y) \to 0$, there exists some absolute constant $C_\epsilon > 0$ such that*

$$(x - p)^\top M (x - p) \geq C_\epsilon \quad, \forall x \notin \mathcal{N}_\epsilon(p_1, p_2). \tag{18}$$

*Proof.* Since, $\beta_i \geq m(\delta_1 + \delta_2 + \max\{\gamma_1, \gamma_2\})$ for $m \geq 1$, the sum of the first row of $M + M^\top = 4\beta_1 - 2\theta_1 \gamma_1 - (\delta_1 + \delta_2 + \theta_2 \gamma_1 + \theta_1 \gamma_2) > 0$. Similarly, the sum of the second row of $M + M^\top$ is also strictly greater than 0, and hence $M + M^\top$ is a symmetric strict diagonally dominant matrix, and hence positive definite.[17] Therefore, for any $x \in \mathbb{R}^2$ and $x \neq 0$, we have $x^\top (M + M^\top) x > 0$. Since $x^\top M x = x^\top M^\top x$, this implies our desired result $x^\top M x > 0$

We now show Equation (18). The reciprocity condition implies that there exists some $\eta_\epsilon > 0$ such that when $\|x - p\| < \eta_\epsilon$, it holds that $\sum_{i=1,2} D_i(x_i, p_i) < \epsilon$. Hence, for any $x \notin \mathcal{N}_\epsilon(p_1, p_2)$, it must be the case that $\|x - p\| \geq \eta_\epsilon$. So, for $x \notin \mathcal{N}_\epsilon(p_1, p_2)$, we have

$$
\begin{aligned}
(x - p)^\top M (x - p) &= \frac{1}{2}(x - p)^\top (M + M^\top)(x - p) \\
&\geq \frac{1}{2}\lambda_{\min}(M + M^\top)\|x - p\|^2 \\
&\geq \frac{1}{2}\lambda_{\min}(M + M^\top)\eta_\epsilon^2 := C_\epsilon,
\end{aligned}
$$

where $\lambda_{\min}(M + M^\top)$ is the minimum eigenvalue of the matrix $M + M^\top$ (which is positive due to the fact that $M + M^\top$ is positive definite). $\qquad\square$

Now, returning to the proof for Theorem 5.1.

**Step 1: show $(p_{1,t}, p_{2,t})$ must enter an $\epsilon$-neigborhood of the SNE prices $(p_1^*, p_2^*)$ infinitely many times.** We use a contradiction argument. Fix any $\epsilon > 0$ and let $C_\epsilon > 0$ be the absolute constant defined in Lemma D.1. Assume by contradiction that $(p_{1,t}, p_{2,t})$ only visits $\mathcal{N}_\epsilon(p_1^*, p_2^*) = \{p \in \mathbb{R}^2 : \|p - (p_1^*, p_2^*)\| < \epsilon\}$ finitely many times, i.e. there exists some $t_\mathcal{N}$ s.t. $\|(p_{1,t}, p_{2,t}) - (p_1^*, p_2^*)\| \geq \epsilon$ for all $t \geq t_\mathcal{N}$. Further, in the first part of this theorem we showed that $\sum_{i=1,2}\theta_i p_{i,t} \to r_t$, so without loss of generality (by taking $t_\mathcal{N}$ large enough), we can also assume for some small $\eta > 0$ such that $2\eta\gamma(\bar{p} - \underline{p}) < C_\epsilon$ (where $\gamma = \max\{\gamma_1, \gamma_2\}$), we have $\left| r_t - \sum_{i=1,2}\theta_i p_{i,t} \right| \leq \eta$ for all $t \geq t_\mathcal{N}$.

We start by deducing a recurrence relationship between $D_i(p_i^*, p_{i,t+1})$ and $D_i(p_i^*, p_{i,t})$ as followed:

$$
\begin{aligned}
&D_i(p_i^*, p_{i,t+1}) \\
&\overset{(a)}{\leq} D_i(p_i^*, p_{i,t}) - \epsilon_{i,t}(g_i^* - g_{i,t})(p_i^* - p_{i,t}) + \frac{(\epsilon_{i,t})^2 g_{i,t}^2}{2\sigma_i} \\
&= D_i(p_i^*, p_{i,t}) - \epsilon_{i,t}\left(2\beta_i p_i^* - \delta_i p_{-i}^* + \gamma_i r^* - 2\beta_i p_{i,t} - \delta_i p_{-i,t} + \gamma_i r_t\right)(p_i^* - p_{i,t}) + \frac{(\epsilon_{i,t})^2 g_{i,t}^2}{2\sigma_i} \\
&\overset{(b)}{\leq} D_i(p_i^*, p_{i,t}) \\
&\quad - \epsilon_{i,t}\left(2\beta_i p_i^* - \delta_i p_{-i}^* + \gamma_i\left(\sum_{j=1,2}\theta_j p_j^*\right) - 2\beta_i p_{i,t} - \delta_i p_{-i,t} + \gamma_i\left(\sum_{j=1,2}\theta_j p_{j,t}\right)\right)(p_i^* - p_{i,t}) \\
&\quad + \epsilon_{i,t}\eta\gamma_i(\bar{p} - \underline{p}) + \frac{(\epsilon_{i,t})^2 g_{i,t}^2}{2\sigma_i} \\
&= D_i(p_i^*, p_{i,t}) - \epsilon_{i,t}\left((2\beta_i - \theta_i\gamma_i)(p_i^* - p_{i,t}) - (\delta_i + \theta_{-i}\gamma_i)(p_{-i}^* - p_{-i,t})\right)(p_i^* - p_{i,t}) \\
&\quad + \epsilon_{i,t}\eta\gamma_i(\bar{p} - \underline{p}) + \frac{(\epsilon_{i,t})^2 g_{i,t}^2}{2\sigma_i},
\end{aligned}
$$

(19)

where in (a) we directly evoked Corollary E.1.1; and in (b) we used the fact that $r^* = \sum_{j=1,2}\theta_j p_j^*$ according to the definition of the SNE, and $\left| r_t - \sum_{j=1,2}\theta_j p_{j,t} \right| \leq \eta$ for $t \geq t_\mathcal{N}$. Summing the

above over $i = 1, 2$ and recalling $\epsilon_{i,t} = \epsilon_t$ we have

$$
\begin{aligned}
& \sum_{i=1,2} D_i(p_i^*, p_{i,t+1}) \\
& \overset{(a)}{\leq} \sum_{i=1,2} D_i(p_i^*, p_{i,t}) - \epsilon_t \left(\boldsymbol{p}^* - \boldsymbol{p}_t\right)^\top M \left(\boldsymbol{p}^* - \boldsymbol{p}_t\right) + 2\epsilon_t \eta \gamma \left(\bar{p} - \underline{p}\right) + \frac{\left(\epsilon_t\right)^2 c_2}{\sigma} \\
& \overset{(b)}{\leq} \sum_{i=1,2} D_i(p_i^*, p_{i,t}) - \epsilon_t C_\epsilon + 2\epsilon_t \eta \gamma \left(\bar{p} - \underline{p}\right) + \frac{\left(\epsilon_t\right)^2 c_2}{\sigma} \\
& = \sum_{i=1,2} D_i(p_i^*, p_{i,t}) - \epsilon_t \left(C_\epsilon - 2\eta \gamma \left(\bar{p} - \underline{p}\right)\right) + \frac{\left(\epsilon_t\right)^2 c_2}{\sigma}
\end{aligned}
\tag{20}
$$

where in (a) we take some finite $c_2 > \max_{i \in \{1,2\}, t \in \mathbb{N}^+} g_{i,t}^2$ for all $i, t$ by recalling Equation (5) which states $g_{i,t} = g_i(\boldsymbol{p}_t, r_t) = 2\beta_i p_{i,t} - (\alpha_i + \delta_i p_{-i,t} + \gamma_i r_t)$, and that $p_{i,t}, r_t$ are bounded within $[\underline{p}, \bar{p}]$ for all $i, t$. Also recall $\gamma = \max\{\gamma_1, \gamma_2\}$ and $\sigma = \min\{\sigma_1, \sigma_2\}$. In (b) we applied Lemma D.1 since $\|(p_{1,t}, p_{2,t}) - (p_1^*, p_2^*)\| \geq \epsilon$ for all $t \geq t_\mathcal{N}$.

Telescoping Equation (20) from some large time period $T + 1$ down to $t_\mathcal{N}$ we get

$$
\begin{aligned}
0 & \leq \sum_{i=1,2} D_i(p_i^*, p_{i,T+1}) \\
& \leq \sum_{i=1,2} D_i(p_i^*, p_{i,t_\mathcal{N}}) - \left(C_\epsilon - 2\eta \gamma \left(\bar{p} - \underline{p}\right)\right) \sum_{t=t_\mathcal{N}}^{T} \epsilon_t + \frac{c_2}{\sigma} \sum_{t=t_\mathcal{N}}^{T} \left(\epsilon_t\right)^2 .
\end{aligned}
\tag{21}
$$

Rearranging terms in Equation (21) and dividing both sides by $\sum_{t=t_\mathcal{N}}^{T} \epsilon_t$ we get the following:

$$
\frac{-\sum_{i=1,2} D_i(p_i^*, p_{i,t_\mathcal{N}})}{\sum_{t=t_\mathcal{N}}^{T} \epsilon_t} \leq -\left(C_\epsilon - 2\eta \gamma \left(\bar{p} - \underline{p}\right)\right) + \frac{c_2}{\sigma} \cdot \frac{\sum_{t=t_\mathcal{N}}^{T} \epsilon_t^2}{\sum_{t=t_\mathcal{N}}^{T} \epsilon_t} .
$$

Since $\sum_{i=1,2} D_i(p_i^*, p_{i,t_\mathcal{N}})$ and $t_\mathcal{N}$ are finite, the condition $\lim_{T \to \infty} \sum_{t=1}^{T} \epsilon_t = \infty$ implies that $\lim_{T \to \infty} \sum_{t=t_\mathcal{N}}^{T} \epsilon_t = \infty$, and the condition $\lim_{T \to \infty} \sum_{t=1}^{T} \epsilon_t^2 < \infty$ implies that $\lim_{T \to \infty} \sum_{t=t_\mathcal{N}}^{T} \epsilon_t^2 < \infty$. Hence, $\frac{\sum_{t=t_\mathcal{N}}^{T} \epsilon_t^2}{\sum_{t=t_\mathcal{N}}^{T} \epsilon_t} = 0$ as $T \to \infty$. Finally, because $C_\epsilon - 2\eta \gamma \left(\bar{p} - \underline{p}\right) > 0$ due to our definition of $\eta$, as $T \to \infty$, the above left hand side goes to zero and the right hand side goes to $-\left(C_\epsilon - 2\eta \gamma \left(\bar{p} - \underline{p}\right)\right) < 0$. This is a contradiction, implying that $(p_{1,t}, p_{2,t})$ must enter an $\mathcal{N}_\epsilon(p_1^*, p_2^*)$ infinitely many times.

**Step 2: show when $(p_{1,t}, p_{2,t})$ enters an $\epsilon$-neigborhood of the SNE prices $(p_1^*, p_2^*)$ for large $t$ (small step size), it must stay in the neighborhood.** Here, we will show that for any $\epsilon > 0$, if $(p_{1,t}, p_{2,t}) \in \mathcal{N}_\epsilon(p_1^*, p_2^*)$ for some large $t$, then $(p_{1,\tau}, p_{2,\tau}) \in \mathcal{N}_\epsilon(p_1^*, p_2^*)$ for all $\tau \geq t$. In fact, we only need to show that $(p_{1,t}, p_{2,t}) \in \mathcal{N}_\epsilon(p_1^*, p_2^*)$ implies $(p_{1,t+1}, p_{2,t+1}) \in \mathcal{N}_\epsilon(p_1^*, p_2^*)$ for large $t$, and the rest follows from an induction argument.

Consider two scenarios, namely $(p_{1,t}, p_{2,t}) \in \mathcal{N}_{\frac{\epsilon}{2}}(p_1^*, p_2^*)$ and $(p_{1,t}, p_{2,t}) \in \mathcal{N}_\epsilon(p_1^*, p_2^*) / \mathcal{N}_{\frac{\epsilon}{2}}(p_1^*, p_2^*)$.

*Scenario 1:* If $(p_{1,t}, p_{2,t}) \in \mathcal{N}_{\frac{\epsilon}{2}}(p_1^*, p_2^*)$, consider some $t_\eta > 0$ such that when $t \geq t_\eta$, we have $\left| r_t - \sum_{i=1,2} \theta_i p_{i,t} \right| \leq \eta$ for some small $\eta$ that satisfies $\frac{\epsilon}{2} > 2\eta \gamma \left(\bar{p} - \underline{p}\right)$. Following the same

deduction as in Equation (20), we have

$$\sum_{i=1,2} D_i(p_i^*, p_{i,t+1}) \;\leq\; \sum_{i=1,2} D_i(p_i^*, p_{i,t}) - \epsilon_t \left(\boldsymbol{p}^* - \boldsymbol{p}_t\right)^\top M \left(\boldsymbol{p}^* - \boldsymbol{p}_t\right) + 2\epsilon_t \eta\gamma \left(\bar{p} - \underline{p}\right) + \frac{(\epsilon_t)^2 c_2}{\sigma}$$

$$\overset{(a)}{\leq} \; \frac{\epsilon}{2} + 2\epsilon_t \eta\gamma \left(\bar{p} - \underline{p}\right) + \frac{(\epsilon_t)^2 c_2}{\sigma}$$

$$= \; \frac{\epsilon}{2} + \epsilon_t \left(2\eta\gamma \left(\bar{p} - \underline{p}\right) + \frac{\epsilon_t c_2}{\sigma}\right)$$

$$\overset{(b)}{\leq} \; \epsilon \,.$$

(22)

In (a) we used Lemma D.1 such that $\left(\boldsymbol{p}^* - \boldsymbol{p}_t\right)^\top M \left(\boldsymbol{p}^* - \boldsymbol{p}_t\right) \geq 0$, and the fact that $(p_{1,t}, p_{2,t}) \in \mathcal{N}_{\frac{\epsilon}{2}}(p_1^*, p_2^*)$ so by definition of an $\epsilon$-neighborhood (see Lemma D.1) $\sum_{i=1,2} D_i(p_i^*, p_{i,t+1}) \leq \frac{\epsilon}{2}$. In (b), we considered large $t$ such that $\epsilon_t < \min\left\{1, \frac{\sigma}{c_2}\left(\frac{\epsilon}{2} - 2\eta\gamma\left(\bar{p} - \underline{p}\right)\right)\right\}$ and used the definition of $\eta$ such that $\frac{\epsilon}{2} > 2\eta\gamma\left(\bar{p} - \underline{p}\right)$.

*Scenario 2:* If $(p_{1,t}, p_{2,t}) \in \mathcal{N}_\epsilon(p_1^*, p_2^*)/\mathcal{N}_{\frac{\epsilon}{2}}(p_1^*, p_2^*)$, let $C_{\frac{\epsilon}{2}}$ be defined as in Lemma D.1. Consider some $t'_\eta > 0$ such that when $t \geq t'_\eta$, we have $\left|r_t - \sum_{i=1,2}\theta_i p_{i,t}\right| \leq \eta$ for some small $\eta$ that satisfies $C_{\frac{\epsilon}{2}} > 2\eta\gamma\left(\bar{p} - \underline{p}\right)$. Following the same deduction as in Equation (20), we have

$$\sum_{i=1,2} D_i(p_i^*, p_{i,t+1}) \;\leq\; \sum_{i=1,2} D_i(p_i^*, p_{i,t}) - \epsilon_t\left(C_{\frac{\epsilon}{2}} - 2\eta\gamma\left(\bar{p} - \underline{p}\right)\right) + \frac{(\epsilon_t)^2 c_2}{\sigma}$$

$$\leq \; \epsilon - \epsilon_t\left(C_{\frac{\epsilon}{2}} - 2\eta\gamma\left(\bar{p} - \underline{p}\right)\right) + \frac{(\epsilon_t)^2 c_2}{\sigma}\,,$$

where the final inequality follows from $(p_{1,t}, p_{2,t}) \in \mathcal{N}_\epsilon(p_1^*, p_2^*)$. Taking large $t$ such that $\epsilon_t < \frac{\sigma}{c_2}\left(C_{\frac{\epsilon}{2}} - 2\eta\gamma\left(\bar{p} - \underline{p}\right)\right)$ we get $\sum_{i=1,2} D_i(p_i^*, p_{i,t+1}) \leq \epsilon$.

Combining the above two scenarios, we showed that for any $t > 0$ such that $t \geq \max\{t_\eta, t'_\eta\}$ and $\epsilon_t \leq \min\left\{1, \frac{\sigma}{c_2}\left(\frac{\epsilon}{2} - 2\eta\gamma\left(\bar{p} - \underline{p}\right)\right), \frac{\sigma}{c_2}\left(C_{\frac{\epsilon}{2}} - 2\eta\gamma\left(\bar{p} - \underline{p}\right)\right)\right\}$, $(p_{1,t}, p_{2,t}) \in \mathcal{N}_\epsilon(p_1^*, p_2^*)$ implies $(p_{1,t+1}, p_{2,t+1}) \in \mathcal{N}_\epsilon(p_1^*, p_2^*)$ and hence $(p_{1,\tau}, p_{2,\tau}) \in \mathcal{N}_\epsilon(p_1^*, p_2^*)$ for all large enough $\tau$ (by induction).

### D.3 Proof of Theorem 5.2

Following the same deduction in Equation (19) we have

$$D_i(p_i^*, p_{i,t+1}) \;\leq\; D_i(p_i^*, p_{i,t}) - \epsilon_{i,t}\left(2\beta_i\left(p_i^* - p_{i,t}\right)^2 - \delta_i\left(p_{-i}^* - p_{-i,t}\right)\left(p_i^* - p_{i,t}\right)\right.$$

$$\left. - \gamma_i\left(r^* - r_t\right)\left(p_i^* - p_{i,t}\right)\right) + \frac{(\epsilon_{i,t}g_{i,t})^2}{2\sigma_i}$$

$$\overset{(a)}{\leq} \; D_i(p_i^*, p_{i,t}) - \epsilon_{i,t}\left(\frac{4\beta_i - \delta_i}{2}\left(p_i^* - p_{i,t}\right)^2 - \frac{\delta_i}{2}\left(p_{-i}^* - p_{-i,t}\right)^2\right)$$

$$+ \epsilon_{i,t}\gamma_i\left(r^* - r_t\right)\left(p_i^* - p_{i,t}\right) + \frac{(\epsilon_{i,t}g_{i,t})^2}{2\sigma_i}\,.$$

where in (a) we used the basic inequality $AB \leq (A^2 + B^2)/2$ for $A = p_{-i}^* - p_{-i,t}$ and $B = p_i^* - p_{i,t}$. Now, consider the step-size sequences $\{\epsilon_{i,t}\}_t$ that satisfy

$$\frac{1}{t+1}\cdot\frac{10}{4\beta_i - \delta_i} \;\leq\; \epsilon_{i,t} \;\leq\; \frac{1}{t+1}\cdot\frac{2}{\max\{\delta_i, \gamma_i\}}, \quad i = 1, 2\,.$$

(23)

Equation (23) holds due to the fact that $\beta_i > m(\delta_i + \gamma_i) > 0$ and $m \geq 2$, which further implies $2(4\beta_i - \delta_i) > 8m(\delta_i + \gamma_i) - 2\delta_i > 10(\delta_i + \gamma_i) > 10\max\{\delta_i, \gamma_i\}$. This leads to

$$D_i(p_i^*, p_{i,t+1})$$

$$\leq D_i(p_i^*, p_{i,t}) - \frac{5}{t+1}(p_i^* - p_{i,t})^2 + \frac{1}{t+1}(p_{-i}^* - p_{-i,t})^2 + \frac{2}{t+1}(r^* - r_t)(p_i^* - p_{i,t}) + \frac{(\epsilon_{i,t}g_{i,t})^2}{2\sigma_i}$$

$$\overset{(a)}{\leq} D_i(p_i^*, p_{i,t}) - \frac{5}{t+1}(p_i^* - p_{i,t})^2 + \frac{1}{t+1}(p_{-i}^* - p_{-i,t})^2 + \frac{2}{t+1}(r^* - r_t)(p_i^* - p_{i,t}) + \frac{c_2}{2(t+1)^2},$$

where in (a) we take some $c_2 > \max_{i \in \{1,2\}, t \in \mathbb{N}^+} \frac{4g_{i,t}^2}{\sigma_i \max\{\delta_i, \gamma_i\}^2}$ for all $i, t$ by using the fact that $p_{i,t}, r_t \in \mathcal{P}$. Summing across $i = 1, 2$, we have

$$\sum_{i=1,2} D_i(p_i^*, p_{i,t+1})$$

$$\leq \sum_{i=1,2} D_i(p_i^*, p_{i,t}) - \frac{4}{t+1}\|\boldsymbol{p}^* - \boldsymbol{p}_t\|^2 + \frac{2}{t+1}(r^* - r_t)(p_1^* - p_{1,t} + p_2^* - p_{2,t}) + \frac{c_2}{(t+1)^2}$$

$$\overset{(a)}{\leq} \sum_{i=1,2} D_i(p_i^*, p_{i,t}) - \frac{2}{t+1}\|\boldsymbol{p}^* - \boldsymbol{p}_t\|^2 + \frac{1}{t+1}(r^* - r_t)^2 + \frac{c_2}{(t+1)^2},$$

where in inequality (a) we applied $C(A + B) \leq \frac{C^2}{2} + \frac{(A+B)^2}{2} \leq \frac{C^2}{2} + A^2 + B^2$ for $A = p_1^* - p_{1,t}$, $B = p_2^* - p_{2,t}$ and $C = r^* - r_t$.

When $R_i(z) = z^2$, we have $D_i(p, p') = (p - p')^2$. Therefore, denoting $x_t = \sum_{i=1,2} D_i(p_i^*, p_{i,t}) = \|\boldsymbol{p}^* - \boldsymbol{p}_t\|^2$ for $i = 1, 2$ and $x_{n,t} = (r^* - r_t)^2$, the equation above yields

$$x_{t+1} \leq \left(1 - \frac{2}{t+1}\right)x_t + \frac{1}{t+1}x_{n,t} + \frac{c_2}{(t+1)^2}. \tag{24}$$

We will show via induction that $x_t \leq \frac{c}{t}$ for some $c > 0$ and any $t \in \mathbb{N}^+$. The proof is constructive and will rely on the following definitions, whose motivations will later be clear.

Fix $\rho_a = \left\lceil \frac{a}{1-a} \right\rceil + 1$, $t_a = \left\lceil \frac{\frac{a}{1-a}(\rho_a + 1)}{\rho_a - \frac{a}{1-a}} \right\rceil$, and take any $\bar{\theta}$ such that $\max\{\theta_1, \theta_2\} < \bar{\theta} < 1$. Here, $\lceil x \rceil = \min\{y \in \mathbb{N}^+ : y \geq x\}$ for any $x \in \mathbb{R}$. Note that $\rho_a$ is bounded as $a$ is bounded away from 1. Next, define

$$t_\theta := \min\left\{\tau \in \mathbb{N}^+ : \tau \geq \rho_a \text{ and } \frac{(\rho_a + 1)\log(\tau - \rho_a - 1)}{\tau} \leq \frac{\bar{\theta}}{\max\{\theta_1, \theta_2\}} - 1\right\} \tag{25}$$

$$u := (1 - a)\max\{\theta_1, \theta_2\} \sum_{\tau=1}^{\rho_a + t_a - 1} \frac{a^{-\tau}}{\tau}. \tag{26}$$

Note that $t_\theta$ is bounded because $\max\{\theta_1, \theta_2\}$ is bounded away from one. Further, since $\rho_a$ and $\bar{\theta}$ are constant, and $\log(t) = o(t)$, it is easy to see that $t_\theta$ exists. Furthermore, define

$$\widetilde{t} := \min\left\{\tau > \max\{\rho_a + t_a, t_\theta\} : \right.$$

$$\left. (t - \rho_a) \cdot \left(2(\bar{p} - \underline{p})^2 + u \cdot \frac{2t \cdot (\bar{p} - \underline{p})^2 + c_2 + 1}{1 - \bar{\theta}}\right) < a^{-t} \quad, \text{for } \forall t \geq \tau\right\},$$

$$c := \frac{2\widetilde{t}(\bar{p} - \underline{p})^2 + c_2 + 1}{1 - \bar{\theta}}.$$

Note that $\widetilde{t}$ must exist because the left hand side is quadratic in $t$, while the right hand side is exponential in $t$ for $a \in (0, 1)$. We provide an illustration for the size of $c$ w.r.t. memory parameter $a$ and $\max\{\theta_1, \theta_2\}$ in Figure 2b of Appendix D.7.

Note that the definition of $\widetilde{t}$ and $c$ implies that the following three equations hold

$$a^t \left(2(\bar{p} - \underline{p})^2 + uc\right) < \frac{1}{t - \rho_a} \quad \forall t \geq \widetilde{t} \tag{27}$$

$$2\widetilde{t}(\bar{p} - \underline{p})^2 + c_2 + 1 + \bar{\theta}c = c \tag{28}$$

$$c > 2\widetilde{t}(\bar{p} - \underline{p})^2 . \tag{29}$$

Here, Equation (27) is due to the following: plugging the definition of $c$ into that of $\widetilde{t}$ we get $(\widetilde{t} - \rho_a) \cdot \left(2(\bar{p} - \underline{p})^2 + uc\right) < a^{-\widetilde{t}}$, and since $c = \frac{2\widetilde{t}(\bar{p}-\underline{p})^2 + c_2 + 1}{1 - \bar{\theta}} \leq \frac{2t \cdot (\bar{p}-\underline{p})^2 + c_2 + 1}{1 - \bar{\theta}}$ for any $t \geq \widetilde{t} > \rho_a$, we have

$$(t - \rho_a)\left(2(\bar{p} - \underline{p})^2 + uc\right) \leq (t - \rho_a)\left(2(\bar{p} - \underline{p})^2 + u \cdot \frac{2t \cdot (\bar{p} - \underline{p})^2 + c_2 + 1}{1 - \bar{\theta}}\right) \overset{(a)}{<} a^{-t},$$

where (a) follows from the definition of $\widetilde{t}$. Equation (28) directly follows from the definition of $c$. Equation (29) follows because $\bar{\theta} \in (0, 1)$ and hence $c = \frac{2\widetilde{t}(\bar{p}-\underline{p})^2 + c_2 + 1}{1 - \bar{\theta}} > 2\widetilde{t}(\bar{p}-\underline{p})^2 + c_2 + 1 > 2\widetilde{t}(\bar{p}-\underline{p})^2$. Hence, this implies that $x_t \leq 2(\bar{p} - \underline{p})^2 < \frac{c}{t}$ for any $t = 1 \ldots \widetilde{t}$, where we recall $x_t = \|\boldsymbol{p}^* - \boldsymbol{p}_t\|^2$.

Consider $t \geq \widetilde{t}$. We will now show via induction that $x_{t+1} \leq c/(t+1)$ using our induction hypothesis that $x_\tau \leq c/\tau$ holds for all $\tau = 1, \ldots, t$. Note that the base case $x_t \leq 2(\bar{p} - \underline{p})^2 < \frac{c}{t}$ for any $t = 1 \ldots \widetilde{t}$ is trivially true as we just discussed. Then, multiplying $t(t+1)$ on both sides of the recurrence relation in Equation (24) and telescoping from $\widetilde{t}$ to $t$, we have

$$\begin{aligned}
t(t+1)x_{t+1} &\leq (t-1)tx_t + tx_{n,t} + c_2 \\
&\leq (t-2)(t-1)x_{t-1} + \sum_{\tau=t-1}^{t} tx_{n,\tau} + 2c_2 \\
&\quad \vdots \\
&\leq (\widetilde{t}-1)\widetilde{t} \cdot x_{\widetilde{t}} + \sum_{\tau=\widetilde{t}}^{t} \tau x_{n,\tau} + (t - \widetilde{t} + 1)c_2 \\
&\leq (\widetilde{t}-1)\widetilde{t} \cdot x_{\widetilde{t}} + \sum_{\tau=\widetilde{t}}^{t} \tau x_{n,\tau} + tc_2 .
\end{aligned} \tag{30}$$

We will now bound $x_{n,\tau}$ for all $\tau = \widetilde{t} \ldots t$. Using the definition $r^* = \theta_1 p_1^* + \theta_2 p_2^*$, we get

$$\begin{aligned}
r^* - r_{\tau+1} &= r^* - ar_\tau - (1-a)(\theta_1 p_{1,\tau} + \theta_2 p_{2,\tau}) \\
&= a(r^* - r_\tau) - (1-a)(\theta_1(p_1^* - p_{1,\tau}) + \theta_2(p_2^* - p_{2,\tau})) .
\end{aligned}$$

By convexity, we further have for any $\tau = 1 \ldots t$,

$$\begin{aligned}
x_{n,\tau+1} = (r^* - r_{\tau+1})^2 &\leq ax_{n,\tau} + (1-a)(\theta_1(p_1^* - p_{1,\tau}) + \theta_2(p_2^* - p_{2,\tau}))^2 \\
&\leq ax_{n,\tau} + (1-a)\left(\theta_1(p_1^* - p_{1,\tau})^2 + \theta_2(p_2^* - p_{2,\tau})^2\right) \\
&\leq ax_{n,\tau} + (1-a)\max\{\theta_1, \theta_2\}x_\tau \\
&\overset{(a)}{\leq} ax_{n,\tau} + (1-a)\max\{\theta_1, \theta_2\}\frac{c}{\tau} ,
\end{aligned}$$

where (a) follows from the induction hypothesis, i.e., $x_\tau \le c/\tau$ holds for all $\tau = 1\ldots t$. Using a telescoping argument, we then have for any $t \ge \tilde{t}$,

$$
\begin{aligned}
x_{n,t+1} &\le a x_{n,t} + (1-a)\max\{\theta_1,\theta_2\}\frac{c}{t} \\[2mm]
&\le a^2 x_{n,t-1} + (1-a)c\max\{\theta_1,\theta_2\} \sum_{\tau=t-1}^{t} \frac{a^{t-\tau}}{\tau} \\
&\;\;\vdots \\
&\le a^t x_{n,1} + (1-a)c\max\{\theta_1,\theta_2\} \sum_{\tau=1}^{t} \frac{a^{t-\tau}}{\tau} \\
&= a^t x_{n,1} + (1-a)a^t c\max\{\theta_1,\theta_2\} \left( \sum_{\tau=1}^{\rho_a+t_a-1} \frac{a^{-\tau}}{\tau} + \sum_{\tau=\rho_a+t_a}^{t} \frac{a^{-\tau}}{\tau} \right) \\
&\overset{(a)}{=} a^t \left( x_{n,1} + uc \right) + (1-a)a^t c\max\{\theta_1,\theta_2\} \sum_{\tau=\rho_a+t_a}^{t} \frac{a^{-\tau}}{\tau} \\
&\le a^t \left( 2(\bar{p}-\underline{p})^2 + uc \right) + (1-a)a^t c\max\{\theta_1,\theta_2\} \sum_{\tau=\rho_a+t_a}^{t} \frac{a^{-\tau}}{\tau} \\
&\overset{(b)}{\le} \frac{1}{t-\rho_a} + (1-a)a^t c\max\{\theta_1,\theta_2\} \sum_{\tau=\rho_a+t_a}^{t} \frac{a^{-\tau}}{\tau} \\
&\overset{(c)}{\le} \frac{1+\max\{\theta_1,\theta_2\}c}{t-\rho_a}.
\end{aligned}
$$

Here, (a) follows from the definition of $u$ in Equation (26); (b) follows from Equation (27); and (c) follows from Lemma E.3 since $t \ge \tilde{t} \ge \rho_a + t_a$. Applying this upper bound on $x_{n,t}$ in Equation (30) we have

$$
\begin{aligned}
t(t+1)&x_{t+1} \\
&\le (\tilde{t}-1)\tilde{t} \cdot x_{\tilde{t}} + (1+\max\{\theta_1,\theta_2\}c) \sum_{\tau=\tilde{t}}^{t} \frac{\tau}{\tau-\rho_a-1} + tc_2 \\
&= (\tilde{t}-1)\tilde{t} \cdot x_{\tilde{t}} + (1+\max\{\theta_1,\theta_2\}c) \sum_{\tau=\tilde{t}}^{t} \left( 1 + \frac{\rho_a+1}{\tau-\rho_a-1} \right) + tc_2 \\
&\overset{(a)}{\le} (\tilde{t}-1)\tilde{t} \cdot x_{\tilde{t}} + (1+\max\{\theta_1,\theta_2\}c) \left( t + (\rho_a+1)\log(t-\rho_a-1) \right) + tc_2.
\end{aligned}
$$

where (a) follows from $\sum_{\tau=\tilde{t}}^{t} \frac{1}{\tau-\rho_a-1} \le \int_{\tilde{t}-1}^{t} \frac{1}{\tau-\rho_a-1} d\tau \le \log(t-\rho_a-1)$ since $\tilde{t} \ge \rho_a+3$. Dividing both sides of the above equation by $t(t+1)$, and using the fact that $x_t \le 2(\bar{p}-\underline{p})^2$ for any $t$, we have

$$
\begin{aligned}
x_{t+1} &\le \left( \frac{2(\tilde{t}-1)\tilde{t}(\bar{p}-\underline{p})^2}{t} + c_2 \right) \cdot \frac{1}{t+1} + \frac{1+\max\{\theta_1,\theta_2\}c\left(1+\frac{(\rho_a+1)\log(t-\rho_a-1)}{t}\right)}{t+1} \\
&\overset{(a)}{\le} \frac{2\tilde{t}(\bar{p}-\underline{p})^2 + c_2 + 1 + \max\{\theta_1,\theta_2\}c\left(1+\frac{(\rho_a+1)\log(t-\rho_a-1)}{t}\right)}{t+1} \\
&\overset{(b)}{\le} \frac{2\tilde{t}(\bar{p}-\underline{p})^2 + c_2 + 1 + \bar{\theta}c}{t+1} \\
&\overset{(c)}{=} \frac{c}{t+1}.
\end{aligned}
$$

Here, (a) follows from the fact that $t \geq \widetilde{t}$, so $\frac{2(\widetilde{t}-1)\widetilde{t}(\bar{p}-\underline{p})^2}{t} + c_2 \leq 2\widetilde{t}(\bar{p} - \underline{p})^2 + c_2$; (b) follows from $t \geq \widetilde{t} \geq t_\theta$ so that $\frac{(\rho_a+1)\log(t-\rho_a-1)}{t} \leq \frac{\bar{\theta}}{\max\{\theta_1,\theta_2\}} - 1$ according to Equation (25); finally, (c) follows from Equation (28).

### D.4 Proof of Theorem 5.3

Here, we first provide a roadmap for the proof. Evoking Corollary E.1.1, we get

$$D_i(p_i^*, p_{i,t+1}) \leq D_i(p_i^*, p_{i,t}) - \epsilon_{i,t}(g_i^* - g_{i,t})(p_i^* - p_{i,t}) + \frac{(\epsilon_{i,t})^2(g_i^* - g_{i,t})^2}{2\sigma_i}. \qquad (31)$$

By bounding the first order term $(g_i^* - g_{i,t})(p_i^* - p_{i,t})$ and the second order term $\frac{(g_i^* - g_{i,t})^2}{2\sigma_i}$, we achieve a recursive relation in the form of

$$\sum_{i=1,2,n} D_i(p_i^*, p_{i,t+1}) \leq \sum_{i=1,2,n} D_i(p_i^*, p_{i,t}) + \sum_{i=1,2,n} \kappa_{i,t} x_{i,t},$$

where we recall the definition $x_{i,t} = (p_i^* - p_{i,t})^2$ for $i = 1, 2, n$ as in the proof of Theorem 5.2, and $\kappa_{i,t}$ is some constant that takes negative values if the conditions in the theorem's statement are satisfied. We then argue if $(\boldsymbol{p}_t, r_t)$ does not converge to the SNE, $\sum_{i=1,2,n} D_i(p_i^*, p_{i,t})$ will be greater than some positive constant $\epsilon > 0$ for all large enough $t$. Combining this with the above recursive relationship, this further implies that the distance between the price profile $(\boldsymbol{p}_t, r_t)$ and the SNE decreases by a positive constant for each period. This will eventually contradict the fact that Bregman divergence is positive.

We start our proof by recalling Equation (5) which states $g_i(\boldsymbol{p}, r) = 2\beta_i p_i - (\alpha_i + \delta_i p_{-i} + \gamma_i r)$. Hence,

$$g_i^* - g_{i,t} = 2\beta_i(p_i^* - p_{i,t}) - \delta_i(p_{-i}^* - p_{-i,t}) - \gamma_i(r^* - r_t).$$

Furthermore, for $i = 1, 2$, we have

$$(g_i^* - g_{i,t})^2 \leq 8\beta_i^2 x_{i,t} + 4\delta_i^2 x_{-i,t} + 4\gamma_i^2 x_{n,t}, \qquad (32)$$

where we used $(A + B + C)^2 \leq 2A^2 + 2(B+C)^2 \leq 2A^2 + 4B^2 + 4C^2$ for $A = 2\beta_i(p_i^* - p_{i,t})$, $B = \delta_i(p_{-i}^* - p_{-i,t})$, and $C = \gamma_i(r^* - r_t)$. Hence, we have

$$D_i(p_i^*, p_{i,t+1})$$

$$\leq D_i(p_i^*, p_{i,t}) - \epsilon_{i,t}\left(2\beta_i x_{i,t} - \delta_i(p_{-i}^* - p_{-i,t})(p_i^* - p_{i,t}) - \gamma_i(r^* - r_t)(p_i^* - p_{i,t})\right) + \frac{(\epsilon_{i,t}g_{i,t})^2}{2\sigma_i}$$

$$\overset{(a)}{\leq} D_i(p_i^*, p_{i,t}) - \epsilon_{i,t}\left(2\beta_i x_{i,t} - \frac{\delta_i}{2}(x_{i,t} + x_{-i,t}) - \frac{\gamma_i}{2}(x_{n,t} + x_{i,t})\right) + \frac{(\epsilon_{i,t}g_{i,t})^2}{2\sigma_i}$$

$$= D_i(p_i^*, p_{i,t}) - \epsilon_{1,t}\left(\frac{4\beta_i - \delta_i - \gamma_i}{2} x_{i,t} - \frac{\delta_i}{2} x_{-i,t} - \frac{\gamma_i}{2} x_{n,t}\right) + \frac{(\epsilon_{i,t}g_{i,t})^2}{2\sigma_i}$$

$$\leq D_i(p_i^*, p_{i,t}) - \left(\frac{(4\beta_i - \delta_i - \gamma_i)\epsilon_{i,t}}{2} - \frac{4\beta_i^2(\epsilon_{i,t})^2}{\sigma_i}\right) x_{i,t}$$

$$+ \left(\frac{\delta_i\epsilon_{i,t}}{2} + \frac{2\delta_i^2(\epsilon_{i,t})^2}{\sigma_i}\right) x_{-i,t} + \left(\frac{\gamma_i\epsilon_{i,t}}{2} + \frac{2\gamma_i^2(\epsilon_{i,t})^2}{\sigma_i}\right) x_{n,t}. \qquad (33)$$

In the inequality (a), we used the basic inequality $AB \leq (A^2 + B^2)/2$ twice, and the last inequality is obtained by invoking Equation (32). Furthermore, we have $g_n^* - g_{n,t} = r^* - r_t - (\theta_1(p_1^* - p_{1,t}) + \theta_2(p_2^* - p_{2,t}))$. Thus,

$$(g_n^* - g_{n,t})^2 \leq \frac{1}{2} x_{n,t} + \frac{1}{2}(\theta_1(p_1^* - p_{1,t}) + \theta_2(p_2^* - p_{2,t}))^2$$

$$\overset{(a)}{\leq} \frac{1}{2} x_{n,t} + \frac{1}{2}(\theta_1 x_{1,t} + \theta_2 x_{2,t}),$$

where (a) follows from $\theta_1 + \theta_2 = 1$ and convexity. By applying the above inequality in Equation (31) with $i = n$, we have

$$D_n(p_n^*, p_{n,t+1})$$

$$\leq D_n(p_n^*, p_{n,t}) - (1-a)\left(\frac{1}{2}x_{n,t} - \frac{\theta_1}{2}x_{1,t} - \frac{\theta_2}{2}x_{2,t}\right) + \frac{((1-a)g_{n,t})^2}{2}$$

$$\leq D_n(p_n^*, p_{n,t}) - \left(\frac{1-a}{2} - \frac{(1-a)^2}{4}\right)x_{n,t}$$

$$+ \left(\frac{(1-a)\theta_1}{2} + \frac{(1-a)^2\theta_1}{4}\right)x_{1,t} + \left(\frac{(1-a)\theta_2}{2} + \frac{(1-a)^2\theta_2}{4}\right)x_{2,t}, \qquad (34)$$

where in the second inequality, we again use the inequality $g_{n,t}^2 \leq \frac{1}{2}x_{n,t} + \frac{1}{2}(\theta_1 x_{1,t} + \theta_2 x_{2,t})$.

Summing up Equations (33) (over $i = 1, 2$) and (34), and collecting terms yields

$$\sum_{i=1,2,n} D_i(p_i^*, p_{i,t+1}) \leq \sum_{i=1,2,n} D_i(p_i^*, p_{i,t}) + \sum_{i=1,2,n} \kappa_{i,t}x_{i,t}, \qquad (35)$$

where the coefficient for $x_{i,t}$ is

$$\kappa_{i,t} = \begin{cases} -\frac{(4\beta_i - \delta_i - \gamma_i)\epsilon_{i,t}}{2} + \frac{4\beta_i^2\epsilon_{i,t}^2}{\sigma_i} + \frac{\delta_{-i}\epsilon_{-i,t}}{2} + \frac{2\delta_{-i}^2\epsilon_{-i,t}^2}{\sigma_{-i}} + \frac{(1-a)\theta_i}{2} + \frac{(1-a)^2\theta_i}{4}, & i = 1, 2 \\ -\frac{1-a}{2} + \frac{(1-a)^2}{4} + \frac{\gamma_1\epsilon_{1,t}}{2} + \frac{2\gamma_1^2\epsilon_{1,t}^2}{\sigma_1} + \frac{\gamma_2\epsilon_{2,t}}{2} + \frac{2\gamma_2^2\epsilon_{2,t}^2}{\sigma_2}, & i = n \end{cases} \qquad (36)$$

Now, for $i = 1, 2$, consider taking step size $\epsilon_{i,t} = \frac{z\sigma_i}{\beta_i}(1-a)$, for some constant $z > 0$ that will be determined later, and denote the corresponding $\kappa_{i,t}$ as $\kappa_i(z)$ (we drop the dependence on time $t$ as step sizes are constant), where for $i = 1, 2$,

$$\kappa_i(z) \overset{(a)}{=} -\frac{(4\beta_i - \delta_i - \gamma_i)z}{2\beta_i}(1-a)\sigma_i + 4z^2(1-a)^2\sigma_i + \frac{z\delta_{-i}}{2\beta_{-i}}(1-a)\sigma_{-i}$$

$$+ \frac{2\delta_{-i}^2 z^2}{\beta_{-i}^2}(1-a)^2\sigma_{-i} + \frac{(1-a)\theta_i}{2} + \frac{(1-a)^2\theta_i}{4}$$

$$\overset{(b)}{\leq} -\frac{(4-\frac{1}{m})z}{2}(1-a)\sigma_i + 4z^2(1-a)\sigma_i + \frac{z}{2m}(1-a)\sigma_{-i} + \frac{2z^2}{m^2}(1-a)\sigma_{-i} + \frac{3(1-a)}{4}$$

$$= (1-a)\left(\left(4\sigma_i + \frac{2\sigma_{-i}}{m^2}\right)z^2 - \left(\left(2 - \frac{1}{2m}\right)\sigma_i - \frac{\sigma_{-i}}{2m}\right)z + \frac{3}{4}\right)$$

$$:= (1-a)f_{i,m}(z). \qquad (37)$$

Here, in (a) we substitute $\epsilon_{i,t} = \frac{z\sigma_i}{\beta_i}(1-a)$ for $i = 1, 2$; in (b) we use the fact that $\theta_i, a \in (0, 1)$ (wich implies $(1-a)^2 \leq 1 - a$) and $\beta_i > m(\delta_i + \gamma_i) > m\max\{\delta_i, \gamma_i\}$.

We follow a similar argument as above and obtain

$$\kappa_n(z) \overset{(a)}{=} -\frac{1-a}{2} + \frac{(1-a)^2}{4} + \frac{z\gamma_1\sigma_1}{2\beta_1}(1-a) + \frac{2z^2\gamma_1^2\sigma_1}{\beta_1^2}(1-a)^2$$

$$+ \frac{z\gamma_2\sigma_2}{2\beta_2}(1-a) + \frac{2z^2\gamma_2^2\sigma_2}{\beta_2^2}(1-a)^2$$

$$\overset{(b)}{\leq} (1-a)\left(-\frac{1}{4} + \frac{z\sigma_1}{2m} + \frac{2z^2\sigma_1}{m^2} + \frac{z\sigma_2}{2m} + \frac{2z^2\sigma_2}{m^2}\right)$$

$$= (1-a)\left(\frac{2}{m^2}(\sigma_1 + \sigma_2)z^2 + \frac{1}{2m}(\sigma_1 + \sigma_2)z - \frac{1}{4}\right)$$

$$:= (1-a)f_{n,m}(z), \qquad (38)$$

where in (a) we substitute $\epsilon_{i,t} = \frac{z\sigma_i}{\beta_i}(1-a)$ for $i = 1, 2$; in (b) we used the fact that $\theta_i, a \in (0, 1)$ and $\beta_i > m(\delta_i + \gamma_i) > m\max\{\delta_i, \gamma_i\}$ for any $i = 1, 2$.

Now, recall the definition $\mathcal{S}_{i,m} = \{z > 0 : f_{i,m}(z) < 0\}$. Then, if we have $\cap_{i=1,2,n}\mathcal{S}_{i,m} \neq \emptyset$, taking any $s \in \cap_{i=1,2,n}\mathcal{S}_{i,m}$ yields $\kappa_i(s) < 0$ for $i = 1, 2, n$. Hence, Equation (35) now becomes

$$\sum_{i=1,2,n} D_i(p_i^*, p_{i,t+1}) \leq \sum_{i=1,2,n} D_i(p_i^*, p_{i,t}) + \sum_{i=1,2,n} \kappa_i(s) x_{i,t}, \quad \kappa_i(s) < 0. \qquad (39)$$

Therefore, we know that

$$\sum_{i=1,2,n} D_i(p_i^*, p_{i,t+1}) < \sum_{i=1,2,n} D_i(p_i^*, p_{i,t}). \qquad (40)$$

Furthermore, by strong convexity,

$$\sum_{i=1,2,n} D_i(p_i^*, p_{i,t}) \geq \sum_{i=1,2,n} \frac{\sigma_i^2}{2} (p_i^* - p_{i,t})^2 \geq \frac{\min_{i=1,2,n} \sigma_i^2}{2} \|p^* - p_t\|.$$

Hence, for any small $\epsilon > 0$, if there exists some $t_\epsilon \in \mathbb{N}^+$ such that $\sum_{i=1,2,n} D_i(p_i^*, p_{i,t_\epsilon}) \leq \frac{\epsilon \cdot \min_{i=1,2,n} \sigma_i^2}{2}$, then by Equation (40), $\sum_{i=1,2,n} D_i(p_i^*, p_{i,t}) \leq \frac{\epsilon \cdot \min_{i=1,2,n} \sigma_i^2}{2}$ for all $t \geq t_\epsilon$, which further implies $\|p^* - p_t\| \leq \epsilon$ for all $t \geq t_\epsilon$. Hence $(p_t, r_t) \overset{t\to\infty}{\longrightarrow} (p^*, r^*)$.

Thus, it remains to show that for any small $\epsilon > 0$, there exists $t_\epsilon > 0$ such that $\sum_{i=1,2,n} D_i(p_i^*, p_{i,t_\epsilon}) < \epsilon$. We will prove this by contradiction. If this is not the case, there exists $\epsilon > 0$, and $\sum_{i=1,2,n} D_i(p_i^*, p_{i,t}) \geq \epsilon$ for all $t \geq 0$. Define $R(z_1, z_2, z_3) = \sum_{i=1,2,3} R_i(z_i)$ for any $z_1, z_2, z_3 \in \mathbb{R}$, and slightly abuse the notation to define $D : \mathbb{R}^3 \times \mathbb{R}^3 \to \mathbb{R}$ as the Bregman divergence with respect to $R$. In the rest of this proof for simplicity we also write $p^* = (p_1^*, p_2^*, r^*)$ and $p_t = (p_{1,t}, p_{2,t}, r_t)$. A simple analysis shows $D(p^*, p_t) = \sum_{i=1,2,n} D_i(p_i^*, p_{i,t})$.

Since $R_i$ is continuously differentiable (by definition of Bregman divergence), $R$ is also continuously differentiable, and hence it is easy to see for any $x, y \in \mathbb{R}^3$ there exists $\delta > 0$ such that

$$D(x, y) < \epsilon, \quad \forall \|x - y\| < \delta.$$

Since we assumed $D(p^*, p_t) = \sum_{i=1,2,n} D_i(p_i^*, p_{i,t}) \geq \epsilon$ for all $t \geq 0$, the above implies $\|p^* - p_t\| \geq \delta$ for all $t \geq 0$. Hence following Equation (39),

$$
\begin{aligned}
D(p^*, p_{t+1}) &\leq D(p^*, p_t) + \sum_{i=1,2,n} \kappa_i(s) \left(p_i^* - p_{i,t}\right)^2 \\
&\leq D(p^*, p_t) + \max_{i=1,2,n} \kappa_i(s) \sum_{i=1,2,n} \left(p_i^* - p_{i,t}\right)^2 \\
&= D(p^*, p_t) + \max_{i=1,2,n} \kappa_i(s) \cdot \|p^* - p_t\|^2 \\
&\overset{(a)}{\leq} D(p^*, p_t) + \delta^2 \max_{i=1,2,n} \kappa_i(s) \\
&\overset{(b)}{\leq} D(p^*, p_1) + t\delta^2 \max_{i=1,2,n} \kappa_i(s),
\end{aligned}
$$

where (a) follows because $\kappa_i(s) < 0$ for $i = 1, 2, n$ and $\|p^* - p_t\| \geq \delta$ for all $t \geq 0$; (b) follows from a telescoping argument. Finally, $\max_{i=1,2,n} \kappa_i(s) < 0$ implies the right hand side in the above inequality goes to negative infinity as $t$ goes to infinity. This implies that $D(p^*, p_{t+1}) = \sum_{i=1,2,n} D_i(p_i^*, p_{i,t}) \leq -\infty$, which contradicts nonnegativity of Bregman divergence. Hence, for any small $\epsilon > 0$, there exists $t_\epsilon > 0$ such that $\sum_{i=1,2,n} D_i(p_i^*, p_{i,t_\epsilon}) < \epsilon$, concluding the proof.

### D.5 Proof of Corollary 5.3.1

When $\sigma_1 = \sigma_2 = \sigma$, Equation (6) becomes

$$f_{i,m}(z) = \begin{cases} 2\sigma\left(2 + \frac{1}{m^2}\right) z^2 - \sigma\left(2 - \frac{1}{m}\right) z + \frac{3}{4} & i = 1, 2 \\ \frac{4\sigma}{m^2} z^2 + \frac{\sigma}{m} z - \frac{1}{4} & i = n \end{cases}.$$

Since in this case the function $f_{1,m}(z)$ and $f_{2,m}(z)$ are identical, we only consider $f_{1,m}(z)$. Note that the function $f_{1,m}(z)$ has two distinct zero roots if and only if its discriminant is strictly greater than 0, i.e., $\left(1 - \frac{1}{2m}\right)^2 - \frac{3}{2\sigma}\left(2 + \frac{1}{m^2}\right) > 0$ which is equivalent to $\sigma > \frac{6(2m^2+1)}{(2m-1)^2}$. Therefore, when $f_{1,m}$ has two distinct zero roots, the smaller one is given by

$$
\begin{aligned}
z_1 &= \frac{1 - \frac{1}{2m} - \sqrt{\left(1 - \frac{1}{2m}\right)^2 - \frac{3}{2\sigma}\left(2 + \frac{1}{m^2}\right)}}{2\left(2 + \frac{1}{m^2}\right)} \\
&= \frac{1}{\sigma} \cdot \underbrace{\frac{3/4}{1 - \frac{1}{2m} + \sqrt{\left(1 - \frac{1}{2m}\right)^2 - \frac{3}{2\sigma}\left(2 + \frac{1}{m^2}\right)}}}_{A} > 0.
\end{aligned}
\tag{41}
$$

Similarly, the discriminant of $f_{n,m}$ (i.e., $\frac{1}{m^2} + \frac{4}{\sigma m^2}$) is always positive, so $f_{n,m}$ always has two zero roots. The larger one is given by

$$
z_2 = \frac{-\frac{1}{m} + \sqrt{\frac{1}{m^2} + \frac{4}{\sigma m^2}}}{\frac{8}{m^2}} = \frac{1}{\sigma} \cdot \underbrace{\frac{1/2}{\frac{1}{m} + \sqrt{\frac{1}{m^2} + \frac{4}{\sigma m^2}}}}_{B} > 0.
\tag{42}
$$

For any $\sigma > \frac{6(2m^2+1)}{(2m-1)^2}$, the two roots of $f_{1,m}(z)$ are both positive, while $f_{n,m}(z)$ always has one positive root and one negative root. Hence, using a simple geometric argument regarding two quadratic functions, it is easy to see that if $z_2 > z_1$, any $s \in (z_1, z_2)$ satisfies $f_{1,m}(s), f_{n,m}(s) < 0$.

Now, consider $z_2 - z_1 = \frac{1}{\sigma}(B - A)$. Since we observe $A$ is decreasing in $\sigma$ and $B$ is increasing in $\sigma$, we have $B - A$ is increasing in $\sigma$. By direct calculations, we see that when $\sigma = \frac{(2m^2+7)^2}{8m^3 - 36m + 8} > 0$, $z_1 = z_2$ (i.e., $B - A = 0$). Therefore because $B - A$ is increasing in $\sigma$, we conclude $B - A > 0$ for any

$$
\sigma > \sigma_0 := \max\left\{\frac{6(2m^2+1)}{(2m-1)^2}, \frac{(2m^2+7)^2}{8m^3 - 36m + 8}\right\},
$$

which implies $z_2 > z_1$ for any $\sigma > \sigma_0$.

In sum, we conclude for any $m > 2$, if $\sigma > \sigma_0$, there exists $s > 0$ that depends on $\sigma$ and $m$ such that $f_{i,m}(s) < 0$ for $i = 1, 2, n$, and by Theorem 5.3, this implies that there exist constant step sizes under which prices and reference prices converge to the unique interior SNE.

### D.6 Proof of Theorem 5.4

When $R_i(x) = \frac{\sigma}{2}x^2$ for $i = 1, 2$, we have $\sigma_1 = \sigma_2 = \sigma$, and Equation (6) becomes

$$
f_{i,m}(z) = \begin{cases} 2\sigma\left(2 + \frac{1}{m^2}\right)z^2 - \sigma\left(2 - \frac{1}{m}\right)z + \frac{3}{4}, & i = 1, 2 \\ \frac{4\sigma}{m^2}z^2 + \frac{\sigma}{m}z - \frac{1}{4}, & i = n \end{cases}.
$$

We define $h_{i,m}(z) := f_{i,m}(z)/2\sigma$ for $i = 1, 2$ and $h_{n,m}(z) := f_{n,m}(z)$, i.e.

$$
h_{i,m}(z) = \begin{cases} \left(2 + \frac{1}{m^2}\right)z^2 - \left(1 - \frac{1}{2m}\right)z + \frac{3}{8\sigma} & i = 1, 2 \\ \frac{4\sigma}{m^2}z^2 + \frac{\sigma}{m}z - \frac{1}{4} & i = n \end{cases}.
\tag{43}
$$

Note that for any $i = 1, 2, n$, $f_{i,m}(z) < 0$ if and only if $h_{i,m}(z) < 0$. Hence, according to Corollary 5.3.1, we know that when $m > 2$ and $\sigma > \sigma_0 = \max\left\{\frac{6(2m^2+1)}{(2m-1)^2}, \frac{(2m^2+7)^2}{8m^3 - 36m + 8}\right\}$, for any $M \in (z_1, z_2)$ (defined in Equations (41) and (42)) we have $f_{i,m}(s) < 0$ for $i = 1, 2, n$, which implies $h_{i,m}(s) < 0$ for $i = 1, 2, n$. Furthermore, via a simple geometric argument, the quadratic functions $h_{1,m}$ (with two positive zero roots) and $h_{n,m}$ (with two zero roots, one positive and one negative) have a unique intersection point $\widetilde{s} \in (z_1, z_2)$. Define $H := h_{1,m}(\widetilde{s}) = h_{2,m}(\widetilde{s}) = h_{n,m}(\widetilde{s}) < 0$. Furthermore, since $\min_{m \geq 0} h_{n,m}(z) = -\frac{1}{4}$, we have

$$
-\frac{1}{4} \leq H = h_{1,m}(\widetilde{s}) = h_{2,m}(\widetilde{s}) = h_{n,m}(\widetilde{s}) < 0.
$$

Now, note that when $R_1(x) = R_2(x) = \frac{\sigma}{2}x^2$, $D_1(p, p') = D_2(p, p') = \frac{\sigma}{2}(p - p')^2$. Also recall $R_n(x) = \frac{1}{2}x^2$, so $D_n(p, p') = \frac{1}{2}(p - p')^2$. Hence, $\sum_{i=1,2,n} D_i(p_i^*, p_{i,t}) = \frac{1}{2}(\sigma x_t + x_{n,t})$, where we define $x_t = \|\boldsymbol{p}^* - \boldsymbol{p}_t\|^2$ and $x_{n,t} = (r^* - r_t)^2$ as in the proof of Theorem 5.3. Hence, by taking $\epsilon_{i,t} = \frac{\sigma \widetilde{s}(1-a)}{\beta_i}$, and continuing from Equation (39), we get

$$\frac{1}{2}(\sigma x_{t+1} + x_{n,t+1}) \leq \frac{1}{2}(\sigma x_t + x_{n,t}) + \sum_{i=1,2,n} \kappa_i(\widetilde{s})x_{i,t}$$

$$\overset{(a)}{\leq} \frac{1}{2}(\sigma x_t + x_{n,t}) + (1 - a)\sum_{i=1,2,n} f_{i,m}(\widetilde{s})x_{i,t}$$

$$\overset{(b)}{=} \frac{1}{2}(\sigma x_t + x_{n,t}) + (1 - a)\left(\sum_{i=1,2} \sigma h_{i,m}(\widetilde{s})x_{i,t} + h_{n,m}(\widetilde{s})x_{n,t}\right)$$

$$\overset{(c)}{=} \frac{1}{2}(\sigma x_t + x_{n,t}) + (1 - a)H \cdot (\sigma x_t + x_{n,t})$$

$$= \frac{1}{2}(1 + 2(1 - a)H)(\sigma x_t + x_{n,t}).$$

Here, (a) follows from upper bounding $\kappa_i(z)$ with $f_{i,m}(z)$ for any $z > 0$ and $i = 1, 2, n$ in Equations (37) and (38) within the proof of Theorem 5.3; (b) follows from the definition of $h_{i,m}$ in Equation (43); (c) follows from the definition of $H := h_{i,m}(\widetilde{s}) \in [-\frac{1}{4}, 0)$ for $i = 1, 2, n$ and $x_t = x_{1,t} + x_{2,t}$.

Using a telescoping argument, we have

$$\sigma x_t < \sigma x_t + x_{n,t} \leq (1 + 2(1 - a)H)^t (\sigma x_1 + x_{n,1}) \leq (\sigma x_1 + x_{n,1})\left(\frac{1 + a}{2}\right)^t,$$

where the final inequality follows from $0 < 1 + 2(1 - a)H \leq \frac{1+a}{2}$ since $H \in [-\frac{1}{4}, 0)$. Finally, because $x_1 \leq 2(\bar{p} - \underline{p})^2$ and $x_{n,1} \leq (\bar{p} - \underline{p})^2$, we have

$$x_t < \left(x_1 + \frac{1}{\sigma}x_{n,1}\right)\left(\frac{1 + a}{2}\right)^t \leq \frac{1 + 2\sigma}{\sigma}(\bar{p} - \underline{p})^2\left(\frac{1 + a}{2}\right)^t.$$

### D.7 Supplementary Figures for Section 5

Figure 2: (a) $\sigma_0$ as a function of sensitivity margin $m$, where $\sigma_0$ is defined in Corollary 5.3.1 (b) Illustration of absolute constant $c$ in Theorem 5.2 w.r.t. memory parameter $a$ and $\max\{\theta_1, \theta_2\}$. All other model parameters take respective values as in Example 1, and firm $i = 1, 2$ again adopts regularizer $R_i(z) = \frac{1}{2}z^2$.

# E Supplementary Lemmas of Section 5

**Lemma E.1.** *For $i = 1, 2, n$ and any $\tilde{z} \in \mathcal{P}$, we have for any $t \in \mathbb{N}^+$,*

$$D_i(\tilde{z}, p_{i,t+1}) \leq D_i(\tilde{z}, p_{i,t}) + \epsilon_{i,t} \cdot g_{i,t} (\tilde{z} - p_{i,t}) + \frac{(\epsilon_{i,t} g_{i,t})^2}{2\sigma_i} . \tag{44}$$

*Proof.* In the projection step of Algorithm 1, we have $p_{i,t+1} = \Pi_{\mathcal{P}}(y_{i,t+1})$. Since we are working with one-dimensional decision sets, it is easy to see that $\Pi_{\mathcal{P}}(y_{i,t+1}) = \arg\min_{p \in \mathcal{P}} D_i(p, y_{i,t+1})$ due to convexity of $R_i$. Recalling the definition $R'_i(p) = \frac{dR_i(z)}{dz}\big|_{z=p}$, we have

$$
\begin{aligned}
p_{i,t+1} = \arg\min_{p \in \mathcal{P}} D_i(p, y_{i,t+1}) &= \arg\min_{p \in \mathcal{P}} R_i(p) - R_i(y_{i,t+1}) - R'_i(y_{i,t+1})(p - y_{i,t+1}) \\
&= \arg\min_{p \in \mathcal{P}} R_i(p) - p \cdot R'_i(y_{i,t+1}) \\
&\overset{(a)}{=} \arg\min_{p \in \mathcal{P}} R_i(p) - p \cdot (R'_i(p_{i,t}) - \epsilon_{i,t} g_{i,t}) \\
&= \arg\min_{p \in \mathcal{P}} R_i(p) - R_i(p_{i,t}) - R'_i(p_{i,t}) (p - p_{i,t}) + p \cdot \epsilon_{i,t} g_{i,t} \\
&= \arg\min_{p \in \mathcal{P}} D_i(p, p_{i,t}) + p \cdot \epsilon_{i,t} g_{i,t} .
\end{aligned}
$$

Here (a) follows from the proxy update step in Algorithm 1. Now, evoking Lemma E.2 (ii) by taking $x = p$, $f(p) = p \cdot \epsilon_{i,t} g_{i,t}$, $z = p_{i,t}$, $y = \tilde{z} \in \mathcal{P}$, we have

$$D_i(\tilde{z}, p_{i,t+1}) \leq D_i(\tilde{z}, p_{i,t}) + \epsilon_{i,t} g_{i,t} (\tilde{z} - p_{i,t+1}) - D_i(p_{i,t}, p_{i,t+1}) .$$

It then follows that

$$
\begin{aligned}
& D_i(\tilde{z}, p_{i,t+1}) \\
\leq\ & D_i(\tilde{z}, p_{i,t}) + \epsilon_{i,t} g_{i,t} (\tilde{z} - p_{i,t}) + \epsilon_{i,t} g_{i,t} (p_{i,t} - p_{i,t+1}) - D_i(p_{i,t}, p_{i,t+1}) \\
\overset{(a)}{\leq}\ & D_i(\tilde{z}, p_{i,t}) + \epsilon_{i,t} g_{i,t} (\tilde{z} - p_{i,t}) + \epsilon_{i,t} g_{i,t} (p_{i,t} - p_{i,t+1}) - \frac{\sigma_i}{2} (p_{i,t} - p_{i,t+1})^2 \\
\leq\ & D_i(\tilde{z}, p_{i,t}) + \epsilon_{i,t} g_{i,t} (\tilde{z} - p_{i,t}) + \frac{(\epsilon_{i,t} g_{i,t})^2}{2\sigma_i} ,
\end{aligned}
$$

where (a) follows from strong convexity of $R_i$. $\qquad\square$

**Corollary E.1.1.** *Under Assumption 1, let $(\boldsymbol{p}^*, r^*)$ be the unique interior SNE as illustrated in Lemma 3.2, then for $i = 1, 2, n$,*

$$g_i^* = \left.\frac{\partial \widetilde{\pi}_i}{\partial p_i}\right|_{\boldsymbol{p}=\boldsymbol{p}^*, r=r^*} = 0 , \tag{45}$$

*and for any $t \in \mathbb{N}^+$*

$$D_i(p_i^*, p_{i,t+1}) \leq D_i(p_i^*, p_{i,t}) - \epsilon_{i,t} (g_i^* - g_{i,t}) (p_i^* - p_{i,t}) + \frac{(\epsilon_{i,t} g_{i,t})^2}{2\sigma_i} . \tag{46}$$

*Proof.* Similar to the proof of Lemma 3.2 and Proposition 4.2, the SNE $(\boldsymbol{p}^*, r^*)$ must satisfy first order conditions w.r.t. quadratic cost function $\widetilde{\pi}_1, \widetilde{\pi}_2, \widetilde{\pi}_n$, respectively, due to the fact that it lies in the interior of the decision set. So $g_i^* = 0$ for $i = 1, 2, n$.

Furthermore, Evoking Lemma E.1 by replacing $z$ with $p_i^*$ and combining $g_i^* = 0$ yields the second part of the proof. $\qquad\square$

**Lemma E.2** (Lemma 3.1 and 3.2 of [15]). *Let $D : \mathcal{C} \times \mathcal{C} \to \mathbb{R}^+$ be the Bregman divergence associated with convex function $R$ on the convex set $\mathcal{C}$: $D(x, y) = R(x) - R(y) - R'(y)(x - y)$ , $\forall x, y \in \mathcal{C}$. Then,*

*(i) For any $x, y, z \in \mathcal{C}$, $D(x, y) + D(y, z) = D(x, z) + (R'(z) - R'(y))(x - y)\rangle$.*

(ii) *Let $f : \mathcal{C} \to \mathbb{R}$ be any convex function and $z \in \mathcal{C}$. If $x^* = \arg\min_{x \in \mathcal{C}} \{f(x) + D(x, z)\}$, then for any $y \in \mathcal{C}$, we have $f(y) + D(y, z) \geq f(x^*) + D(x^*, z) + D(y, x^*)$.*

The proofs for the above lemma are very standard and we will omit them in this paper.

**Lemma E.3.** *Let $a \in (0, 1)$, $\rho_a = \left\lceil \frac{a}{1-a} \right\rceil + 1$, and $t_a = \left\lceil \frac{\frac{a}{1-a}(\rho_a+1)}{\rho_a - \frac{a}{1-a}} \right\rceil$. Then, for any $t \geq \rho_a + t_a$, we have*

$$\sum_{\tau=\rho_a+t_a}^{t} \frac{a^{-\tau}}{\tau} \leq \frac{1}{1-a} \cdot \frac{a^{-t}}{t - \rho_a}.$$

*Proof of Lemma E.3.* We adopt an induction argument with hypothesis $\sum_{\tau=\rho_a+t_a}^{t} \frac{a^{-\tau}}{\tau} \leq \frac{1}{1-a} \cdot \frac{a^{-t}}{t-\rho_a}$. For the base case, consider $t = \rho_a + t_a$. We can easily see $\frac{a^{-(\rho_a+t_a)}}{\rho_a + t_a} < \frac{1}{1-a} \cdot \frac{a^{-(\rho_a+t_a)}}{t_a}$. Now assume that the induction hypothesis holds for some some $t \geq \rho_a + t_a$. We will show $\sum_{\tau=\rho_a+t_a}^{t+1} \frac{a^{-\tau}}{\tau} \leq \frac{1}{1-a} \cdot \frac{a^{-(t+1)}}{t-\rho_a+1}$. We start with

$$\sum_{\tau=\rho_a+t_a}^{t+1} \frac{a^{-\tau}}{\tau} \leq \frac{1}{1-a} \cdot \frac{a^{-t}}{t-\rho_a} + \frac{a^{-(t+1)}}{t+1} = \frac{a^{-(t+1)}}{1-a} \cdot \left( \frac{a}{t-\rho_a} + \frac{1-a}{t+1} \right).$$

Furthermore,

$$\frac{a}{t-\rho_a} + \frac{1-a}{t+1} - \frac{1}{t-\rho_a+1} = a \left( \frac{1}{t-\rho_a} - \frac{1}{t-\rho_a+1} \right) + (1-a) \left( \frac{1}{t+1} - \frac{1}{t-\rho_a+1} \right)$$

$$= \frac{1}{t-\rho_a+1} \left( \frac{a}{t-\rho_a} - \frac{(1-a)\rho_a}{t+1} \right)$$

$$= \frac{1}{t-\rho_a+1} \cdot \frac{(a - \rho_a(1-a))\, t + (1-a)\rho_a^2 + a}{(t-\rho_a)(t+1)}$$

$$\overset{(a)}{\leq} 0,$$

where (a) follows from $\rho_a = \left\lceil \frac{a}{1-a} \right\rceil + 1 > \frac{a}{1-a}$ and the fact that

$$\frac{(1-a)\rho_a^2 + a}{\rho_a(1-a) - a} = \frac{\rho_a^2 + \frac{a}{1-a}}{\rho_a - \frac{a}{1-a}} = \rho_a + \frac{\frac{a}{1-a}(\rho_a+1)}{\rho_a - \frac{a}{1-a}} < \rho_a + t_a \leq t.$$

Therefore, we can conclude that

$$\sum_{\tau=\rho_a+t_a}^{t+1} \frac{a^{-\tau}}{\tau} \leq \frac{1}{1-a} \cdot \frac{a^{-(t+1)}}{t-\rho_a+1},$$

which is the desired result. $\qquad\square$

## Footnotes

[15]Exposure effects in reference price formation refer to considering reference prices as a weighted average of all historical prices, where weights depend on factors such as sales or number of consumers.

[16]Note to run OMD algorithms in [6], agents need to know the length of the time horizon. Such knowledge is not required in our setting.

[17]A symmetric square matrix has real eigenvalues, and a diagonally dominant square matrix has eigenvalues whose real parts are positive. Hence $M + M^\top$ has positive eigenvalues, and is thus positive definite.