[Reviews · NeurIPS 2020]

Review 1

Summary and Contributions: This paper studies a two-firm problem, where each firm tries to set its price on a single product. Consumers are assumed to have a reference price, which evolves linearly over time based on the prices chosen by the firms at each time step. The demand for the product from firm i at times t is assumed to be linear in the current price vector, and the reference price. The paper studies a stable Nash equilibrium, which is one where each firm is setting a best-response price to the reference price and the price of the other firm. It is shown that such an SNE always exists. Then, the authors show that if each firm runs online mirror descent as a pricing scheme then it is possible to converge to SNE under certain conditions.

Strengths: The theoretical model is clear, and the results are clear. The paper is well written.

Weaknesses: I'm unsure to what extent this work is of interest to the NeurIPS community. The model seems very simple. For example, it does not seem super convincing that everything would be linearly-evolving in prices. It's also unclear why firms would observe the demand gradient. Unclear if this model has any connection to real applications.

Correctness: The claims seem to be correct,

Clarity: Mostly, yes. There are some vagueness issues around the assumption m >= 2 and also around Assumption 1. It is hard to figure out from the body of the paper whether Assumption 1 is invoked in Section 5. There are occasional references to "the unique interior equilibrium," but it's not clear why this unique interior equilibrium exists, unless Assumption 1 is used. Edit: After rereading the text right before Assumption 1, it is stated that Assumption 1 is assumed throughout the rest of the paper. I think this should be explicitly highlighted in all theorem statements, or none (right now it's highlighted in Lemma 3.2, but not in later sections where it's arguably more necessary).

Relation to Prior Work: Yes

Reproducibility: Yes

Additional Feedback: This paper is well-written and I found it interesting to read. However, in the end I gave the paper "marginally below," mostly because I am unsure to what extent the paper would appeal to a NeurIPS audience, and because I am unsure how novel the paper is (I am not very familiar with the related literature). From a game-theoretic perspective I don't think the SNE existence and OMD are results are quite interesting enough to clear the bar. Secondly, from a practical perspective I'm somewhat skeptical about the assumption that demand gradients are observed, and similarly that demands evolve linearly in prices seems somewhat unlikely. On a more specific note, I don't find assumption 1 and footnote 7 very convincing. If the SNE occurs on the boundary then you say that the firm might expand the set of prices. But what if the SNE then loses the best-response condition? In general, why is it guaranteed that they can enlarge the set of prices until a solution occurs in the interior? And what if the set of prices needs to be expanded a lot, then the bounds would be seriously violated? Minor comments: "an OMD algorithm has been shown" the OMD algorithm? or OMD algorithms? "has been studied under various equilibria frameworks" equilibrium "In this work, unless stated otherwise, we consider m ≥ 2. In particular, we only use this condition for showing our convergence results in Section 5." These sentences seem contradictory. Are you assuming m >= 2 only in Section 5? Unclear as currently written. In the best response conditions, after stating the inequality, it should say "for all p in the feasible set"


Review 2

Summary and Contributions: The paper studies the problem of price competition between two firms in a repeated setting. The authors consider a setting in which the value consumers have for the firm is endogeneous, and depends on the history of prices used by the firms. The paper study the concept of stable Nash equilibria, which encodes the traditional condition that the firm best respond to each other, but also ensures that prices are stable (the current reference price coincides with the one induced by the current prices). The main contributions are as follows: 1) The authors show existence and uniqueness of a stable Nash equilibrium of the problem 2) The authors show how no-regret dynamics (in this case, online mirror descent) lead to convergence to a stable Nash, and characterize the convergence rate of the dynamics.

Strengths: First, I think the model used by the authors is interesting. In particular, to the best of my knowledge, the idea of reference prices is novel and adds a layer of complexity to the problem. Unlike the standard model where customer values are exogeneous and the demand is only a function of the prices set by the firms, the current model considers the case when past prices define consumer expectations and have an effect on their value and on the demand. This seems to model effects that commonly arise in real-life but are not often taken into account in the CS-econ literature. Second, the paper seems to get the best of both worlds. On the one hand, it assumes agents act in a no-regret manner, which is a much more flexible and permissible model of agent behavior than assuming they are computing (which can be computationally hard) and playing an equilibrium. A plus of this framework is that, on top of being more natural and realistic, it also allows the authors to deal with settings of incomplete information/the players’ update rely on minimum levels of information. On the other hand, even as the agents play simpler no regret strategies, the dynamics converge to a strong concept of equilibrium where agents best respond to each other and prices are stable. From a technical point of view, I think the idea of explaining the behavior of nature as running its own online mirror descent to compute the reference prices, and reducing the 2-player pricing problem with reference prices to a 3-player pricing problem, is interesting. Finally, the authors show a reasonable (1/t) rate of convergence of their dynamics under the right regularization and choice of (decreasing) step size. They provide additional conditions under which taking a constant step size is sufficient for convergence; in this case, the algorithm gains both in simplicity and in faster convergence rates (e^t), which constitutes a nice addition to the results of the paper.

Weaknesses: One of the main weaknesses of the paper is that references prices and demand functions have a simple linear form, which seems fairly unrealistic. However, I think this is fine as these models have been studied in previous work, and as the focus of the paper is in the novelty of having reference prices and in the way they affect the demand. Further, the experiments and Figure 1 do not seem to add much to the theory, and basically show the behavior already stated in the theorems (except for 1c that exhibits an oscillating behavior for a bad choice of constant step size). I think a useful plot that would complement the theory rather than repeating it would compare the theoretical convergence rates to the ones obtained in practice.

Correctness: I have not been able to check the proofs in the paper very carefully, but the proof ideas seem reasonable at a high level.

Clarity: I think the paper is well written and easy to follow. I believe the authors do a good job of motivating the problem they are looking at and explaining the novelty and the technical challenges of their contributions. I also like that, despite the short page limit, the authors spend a good amount of time discussing their algorithms and results, explaining their choices of parameters, and providing examples of when their conditions hold/do not hold.

Relation to Prior Work: The authors position themselves clearly with respect to previous work. The introduction makes the position they take on the information available to the firms and their model of strategic behavior clear compared to previous work. They also provide a discussion of how their proof techniques differ from previous work in Section 5.3

Reproducibility: Yes

Additional Feedback: Post-rebuttal: thanks to the authors for their detailed response! I think Reviewers 1 and 4 bring up valid concerns, and it seems that some of reviewer 4's concerns are still not fully addressed after the rebuttal. I however did not lower my score and still recommend this paper for acceptance, as I think that the study of reference effects in competition is novel to the best of my knowledge, even if the model proposed by the authors is still simple. I also think that the fact that the authors show that no-regret dynamics converge to SNE is interesting, even if these results may not be too surprising. I believe this work to be of interest to the CS-Econ community in general and to the game theory track at NeurIPS.


Review 3

Summary and Contributions: This paper studies a multi-period model: in each period, each of two firms posts a price for each product. The consumers demand for each of the products is linear in both prices and in addition linear in the reference price which captures past prices. Specifically, it is a weighted average of the previous-period reference price and the two previous-period posted prices. The firms do not know the specific demand function and have access only to the derivative of their revenue (a function that maps a price to revenue which is equal to demand times price) which is denoted by g_i(p_i). Note that g_i() depends on the other parameters but the firm views them as constants and is able to feed g_i with a possible choice of p_i and then get the revenue derivative for that price choice. The paper analyzes a natural learning process through which firms can reach market stability which is defined as a steady state (p^*, r^*) satisfying: (i) p_i^* maximizes the period t revenue of firm i given that the other firms posts a price p_{-i}^* in period t and that the reference price in period t is r^*, (ii) if the state in period t is (p^*, r^*) then the reference price period t+1 is r^*. I believe that this is a natural stability condition, and the paper gives a constructive proof that, for every initial reference price r, there exists a learning process that converges monotonically to market stability. However, this learning process relies on knowing which prices maximize revenue. Naively, this would require knowledge of all parameters – the opponent’s posted price, the reference price, and the coefficients of the demand function. The paper then suggests another learning algorithm that relies on carefully predetermined fixed strongly convex function R and “step sizes” epsilon_{i,t} for every firm i and period t. The algorithm sets the p_{i,t+1} so that the difference between R’(p_{i,t+1}) and R’(p_{i,t}) divided by the step size epsilon_{i,t} will be equal to g_i(p_{i,t}) or to the upper or lower bound of the price interval if no interior point satisfies the equality. The paper shows that with the appropriate choice of the R’s and the step sizes, this algorithm converges to market stability where for any t the Euclidean norm of the difference between p_t and p^* is O(1/t). There is the issue of why the proposed algorithm is different in a meaningful way from standard gradient descent algorithms and other results on learning dynamics in games, which the paper tries to resolve in section 5.3. There are also the standard issues of justifying the demand model and explaining whether the proposed learning algorithm captures any realistic aspects (otherwise, why is it interesting?). I am still in favor of acceptance as I believe that the analysis is novel, that the learning community should better understand such learning dynamics, and that the paper can serve as a beneficial step in that direction.

Strengths: See above.

Weaknesses: See above.

Correctness: See above.

Clarity: YES

Relation to Prior Work: See above.

Reproducibility: Yes

Additional Feedback:


Review 4

Summary and Contributions: This work proposes a model for two firms setting prices in a large consumer market where consumer demand is dependent on a memory of historical prices. Deterministic, fixed step-size dynamics are assumed known for the reference price allowing analysis of the market as a 3-player game. Assuming conditions on the parameters of the consumer demand model, reference price dynamics, and access to gradient information, firms employing no-regret algorithms are proven to converge to a stable Nash equilibrium under decaying, and in special cases, fixed step sizes. The reference price is also shown to stabilize. A few experiments are conducted to support the claims.

Strengths: The theoretical analysis is sound and allows a general family of step sizes and strongly convex regularizers for the firms employing online mirror descent. The general setting firms interacting in an environment that changes due to their interactions is interesting.

Weaknesses: I am unconvinced of the significance and relevance of this work to the NeurIPS community. This work analyzes a 3-player game, each player with a 1-d "action" in a deterministic environment, i.e., the parameters of the game are fixed. If the authors assume the standard Euclidean regularizer, it seems as though the proofs in this paper can be mostly reasoned about by considering a linear dynamical system and basic results from variational inequalities (VI). For example given the known reference price dynamics, the Jacobian of the map F for an appropriate VI(F,X) is: J = [2\beta_1, -\delta_1, -\gamma_1] [-\delta_2, 2\beta_2, -\gamma_2] [-\theta_1, -\theta_2, 1]. Given assumptions on the parameters in the paper (line 90 and 113), this matrix is diagonally dominant, therefore, has eigenvalues with positive real part (note updating the system looks like x <-- x - step * (J * x + b) so positive means continuous time convergent). The authors assume the SNE exists in the interior of the set. With this assumption, the SNE is obviously unique because this linear system has a single fixed point. The upper left 2 x 2 can be shown to be positive definite (J+J^T > alpha * I > 0) which means that the appropriate VI that models the two firms is strongly monotone [1]. The solution to a strongly monotone VI is unique and a Nash equilibrium [2]. The point of this is simply to reason that the results are not very surprising. Given the complexity of the proof techniques used in the paper, I would have liked to have seen results that are general to n firms, consider a stochastic setting or unknown reference price dynamics, and/or examine a non-linear model. [1] "Projected Dynamical Systems and Variational Inequalities with Applications"; Nagurney & Zhang; '12 [2] "Convex Optimization, Game Theory, and Variational Inequality Theory"; Scutari, Palomar, Facchinei, Pang; '10 After rebuttal: 1. I see your point that the SNE w.r.t. a moving ref price is more nuanced than an NE w.r.t. a fixed ref price, but the moving ref price moves in an amount proportional to the firms; if the firms converge, then the ref price converges and is effectively fixed. My argument above was not only about uniqueness, the diagonal dominance also suggests convergence. 2. I understand decaying step sizes complicate analysis. Note fixed heterogenous step sizes are handled by VIs: x <-- x - step * D * (J * x + b) = x - step * (D * J * x + D * b) where D is a pos def diagonal matrix that accounts for differences in step sizes between players. Note diagonal dominance still holds for D * J and monotonicity of the upper left 2 x 2 still holds for a range of relative step sizes for the two firms. IMO, viewing the reference price as an OMD player is not a primary contribution. Any linear update rule can be written as gradient descent on some objective. Also, referring to it as OMD is a bit misleading; the ref price updates with vanilla online gradient descent. Maybe I missed a more general analysis. 3. The player loss functions in [4] are not fixed; they may change with time. Also, the view that opponents in a game can be viewed instead as a non-stationary environment is well known; you simply reverse this argument. All in all, this paper presents an analysis of a deterministic, 3-d, linear dynamical system. Complexities arise from the fact that a) the firms may use more general OMD updates (not necessarily OGD) which could make the system nonlinear and b) decaying step sizes while the ref price uses a fixed step size. I'm raising my score to marginally above as it appears the other reviewers are interested in this analysis. I've relayed my comments to the AC and I'm okay with acceptance if the other reviewers are.

Correctness: I did not check all proofs, particularly Theorem 5.3 and on, but I do not doubt the claims. The empirical methodology also appears sound. The claim is repeatedly made that "In the multi-agent online learning literature, the games are stateless and all agents are required to take decreasing step sizes". This is not true or at least misleading. Online learning specifically looks at environments that change over time, possibly adversarially and agents use no-regret algorithms with fixed step sizes (assuming the time horizon is known) [3,4]. Moreover, the most basic algorithm from variational inequality theory, the "projection method", converges for fixed step sizes assuming certain properties of the vector field (e.g., strongly monotone) [1,2]. [3] "No-regret learning in convex games"; Gordon, Greenwald, and Marks; '08 [4] "Online Monotone Games"; Gemp and Mahadevan; '17 *note the player loss functions here are not fixed; they may change every time step.

Clarity: The paper is well written although possibly overly dense given the simplicity of the model. Some of my concern likely comes from not fully digesting that the system being analyzed was linear (regularizers aside) and 3-d. The paper may benefit from providing the reader with more information up front. A paragraph giving a simple proof sketch or explanation near the beginning of the paper outlining why the results are likely true followed by an argument for why an in-depth analysis is both important and challenging is missing. The modeling decisions were generally well motivated however the authors argue parameters are chosen in a way because it is sensible that "demand is most affected by a firm’s own prices relative to competitor’s prices and surcharge/discounts." It is a common technique when designing a linear system to choose parameters such that the system's Jacobian is diagonally dominant so that the system will be convergent. It seems this was likely an additional motivation and would be worth mentioning to the reader if so.

Relation to Prior Work: The related work focuses primarily on the economics literature, which is partly telling why this paper feels less relevant to the NeurIPS community. I am not familiar with many papers in the ML literature similar to this one (e.g., [5] maybe?) and so I'm not sure what previous contributions might be missing. [5] "Smooth markets: A basic mechanism for organizing gradient-based learners; Balduzzi et al; '20

Reproducibility: Yes

Additional Feedback: I will happily raise my score if the other reviewers find this paper relevant and the results important to the "Game Theory and Computational Economics" track as this is my main concern. I realize the proofs in this paper go beyond vanilla "gradient descent" for the firms and the analysis around step sizes is a bit involved -- I don't mean to trivialize your work with my back-of-the-envelope calculations above. It is just to explain that the claims you have proven do not seem like they would be unexpected. I made suggestions above around proving claims for a more complex model. line 40: "has..." --> "have been studied"

[Author Response · NeurIPS 2020]

We first thank all reviewers for their thoughtful comments, and we wish everyone health during these hard times.

**[R1]: 1.** We acknowledge the simplicity in our linear demand and reference price update models. We made these modelling
decisions due to two reasons: (i) We intended to focus on the convergence of prices set by agents (who run mirror descent) when
they may be using step sizes of different orders; e.g. firms use step size $\mathcal{O}(1/\sqrt{t})$ while nature adopts constant step sizes. We
believe that this will serve as a first step to future research on more complex models. (ii) Linear demand and reference price update
models are well studied, and have many practical motivations. For example in marketing, reference prices are used to model
consumer price expectations for particular products, and such expectations are adjusted based on past prices consumers experience;
please see [28] for a comprehensive survey for applications of linear demand models, and [24,45,50] for motivations/empirical
validations for the linear reference price model. These references are also discussed in Section 2 of the paper.

**2.** Regarding practical aspects of gradient feedback, real-world firms typically have a good understanding of price elasticity of
demand due to operational experiences. The gradient of revenue can be calculated using estimated elasticity, observed sales (i.e.
demand) and prices. That being said, we agree that a more practical setting would be to assume firms obtain noisy gradients
or zero-order feedback. Nevertheless, we point out that the main message we hope to convey through this paper is the fact that
firms can converge to the SNE under limited information by running simple yet practical OMD algorithms, which we argue is
a non-trivial result even when firms have exact gradient feedback due to heterogeneity in step sizes (see Section 5.3). We are
definitely interested in generalizing our results to other feedback structures in future work.

**3.** Assumption 1 is invoked in all theorems and lemmas of Section 5, and we will clearly state this in the revised paper. Regarding
footnote 7, consider two different price profiles: $(p, r)$ which is obtained through first order conditions w.r.t. the demand and
reference price update models (see proof of Lemma 3.2 in Appendix B); and the SNE $(p^*, r^*)$, which by definition satisfies the
best-response property. These profiles are identical iff $(p, r)$ lies in the decision set. In the proof of Lemma 3.2, we show that
$(p, r)$ are strictly positive constants given model parameters $\alpha, \beta, \gamma, \delta$ and reference memory parameter $a$. Footnote 7 intends to
say that as long as the decision set lower (upper) bound is small (large) enough while remaining positive, $(p, r)$ will lie in the
decision set. This means if firms are willing to consider both prices near zero and those sufficiently large, Assumption 1 holds.
Nevertheless, we agree that footnote 7 may cause confusion, and we will modify this in the revised paper.

**[R2 & R3]:** We indeed believe the simple linear demand and reference update models capture many realistic aspects of consumer
behavior and firm competition in various markets; please see response 1. for Reviewer #1. Echoing our expanded literature review
in Appendix A, we realized there are not many works in the CS-econ and behavioral economics literature that analyze market
dynamics under reference effects from an online learning perspective. We think it is important to consider demand models which
not only depend on pricing decisions, but also some time-dependent market characteristic (e.g. consumer expectations and choice
behavior). Thus, we view this paper as a first step for analyzing more complex competitions under reference effects, and believe
the learning and game theoretic aspects of this work make it a good fit for NeurIPS. We also thank Reviewer #2 for suggestions
regarding experiments. We are definitely interested in comparing our theoretical convergence rates to those obtained in practice,
despite many practical challenges, e.g. the proprietary nature of data from firms that run similar algorithms. We are certain that this
will be valuable in future extensions of this work. Regarding Reviewer #3's comments on the practical aspects of firms running
OMD, we point out that OMD is an off-the-shelf algorithm that many firms may consider using in practice due to its simplicity. In
this work, we wanted to analyze the impact of running this widely used algorithm in dynamic pricing competitions.

**[R4]: 1.** We agree that uniqueness of SNE can be proved using variational inequalities (VI). However, we would like to point
out that uniqueness, as well as proving uniqueness, is not the focal point of our paper. Our main focus is to understand whether
multiple firms running OMD would imply convergence to the unique SNE under varying reference prices. In fact, our proof
for showing uniqueness under Assumption 1 (Lemma 3.2) is also very straightforward. Moreover, we point out the differences
between an Nash Equilibrium (NE) w.r.t. a fixed reference price, and the SNE. Fixing a reference price, the two firms admit a
unique NE, and VI indeed implies OMD convergence to this particular NE if the reference price remains fixed. However, reference
prices vary according to firms' decisions, so the respective NE changes as firms progress. Hence, the 3-agent system may diverge
and oscillate (Figure 1 (c) gives such an example). Mathematically, in order to use VI to show the update $x \leftarrow x - step * (Jx + b)$
implies convergence, *step* should be identical across all 3 agents (two firms and nature). In our setting, nature takes constant step
sizes, while firms may take decreasing step sizes, so VI is not directly applicable. Furthermore, heterogeneuous step sizes better
model reality, as firms are unaware of how reference prices update and run OMD independently. On a separate note, we agree that
it is beneficial to add discussions that our model admits a diagonally dominant Jacobian, and will revise the paper accordingly.

**2.** As discussed in the introduction and Section 5.3, we are surprised that previous results for multi-agent OMD convergence (e.g.
[9, 33, 41]), whose methodologies rely on variational inequalities, cannot be applied in our setting, primarily because agents (firms
and nature) may be using step sizes of different orders. The fact that convergence still occurs under such heterogeneity is not
intuitive, and we find this phenomena to be interesting. Finally, to the best of our knowledge, our proof techniques for convergence,
as well as the perspective of modeling reference prices as decisions by nature running OMD, are novel.

**3.** Thank you for allowing us to clarify the sentence *"...... games are stateless and all agents are required to take decreasing*
*step sizes"*. When saying the 2-firm game has a "state", we mean agents' utility functions are not only functions of their current
decisions, but also of the time-dependent reference price. We can view our utility functions of interest being parameterized by
varying reference prices. The two works Reviewer #4 pointed out consider each agent optimizing a fixed utility function over
time, and these functions only depend on agents' current decisions. This means the gradient or whatever feedback a firm obtains
after making a decision is with respect to these fixed functions. However, in our case, the gradient feedback can be considered as
associated with "different functions" which are parameterized by each period's reference price. This technicality makes previous
algorithms and convergence results break down in our setting. When we state "all agents are required to take decreasing step
sizes", we were particularly referring to results presented in [9, 33, 41]. Nevertheless, we agree that the sentence pointed out as
well as similar sentences are rather inaccurate, and we will modify the way we explain this in the revised version of the paper.

[Meta-Review · NeurIPS 2020]

There were some concerns about this paper's relevance to NeurIPS, and whether there were any actual applications in practice. But after some discussion we agreed that the paper is suitably interesting and it provides an interesting analysis of convergence to SNE (via no-regret learning) in a two-firm market selling to a family of consumers.